# Deep Learning for BioImaging: What Are We Learning?

**Ivan Svatko**[1,2†], **Maxime Sanchez**[2,3,4,5†], **Ihab Bendidi**[2,6],
**Gilles Cottrell**[1] **& Auguste Genovesio**[2]

[1] Université Paris Cité, IRD, Inserm, MERIT, F-75006, Paris, France
[2] IBENS, Ecole Normale Supérieure, Université PSL, Paris, France
[3] Institut Curie, Université PSL, Paris, France
[4] Iktos, Paris, France
[5] INSERM, U1331, Paris, France
[6] Valence Labs, Recursion, London, United Kingdom
{firstname.lastname}@ens.psl.eu        gilles.cottrell@ird.fr

## Abstract

Representation learning has driven major advances in natural image analysis by enabling models to acquire high-level semantic features. In microscopy imaging, however, it remains unclear what current representation learning methods actually learn. In this work, we conduct a systematic study of representation learning for the two most widely used and broadly available microscopy data types, representing critical scales in biology: cell culture and tissue imaging. To this end, we introduce a set of simple yet revealing baselines on curated benchmarks, including untrained models and simple structural representations of cellular tissue. Our results show that, surprisingly, state-of-the-art methods perform comparably to these baselines. We further show that, in contrast to natural images, existing models fail to consistently acquire high-level, biologically meaningful features. Moreover, we demonstrate that commonly used benchmark metrics are insufficient to assess representation quality and often mask this limitation. In addition, we investigate how detailed comparisons with these benchmarks provide ways to interpret the strengths and weaknesses of models for further improvements. Together, our results suggest that progress in microscopy image representation learning requires not only stronger models, but also more diagnostic benchmarks that measure what is actually learned.

## 1 Introduction

Deep learning has transformed image analysis, supported by a large ecosystem of models, datasets, and benchmarks that make it easier to compare methods and reuse them for downstream applications. A similar shift is now happening in machine learning for biology, especially for microscopy imaging: data volumes are rapidly increasing (Chandrasekaran et al., 2023), and datasets are becoming more diverse in cell types, perturbations, and experimental settings. This scale has enabled the rise of *foundation models* trained on broad microscopy collections (Kenyon-Dean et al., 2025; Bioptimus, 2025), with the promise of transferring to many biological tasks. As the field tests how useful these models really are for downstream biology, benchmarking becomes essential, both to rank models fairly and to understand which capabilities actually transfer. In parallel, new microscopy benchmarking efforts have started to emerge, including tasks focused on cell culture imaging (Chen et al., 2023; Bourriez et al., 2024; Kraus et al., 2024) and histopathology (Jaume et al., 2024; Gindra et al., 2025).

However, systematic benchmarking in microscopy imaging remains limited, and this gap is not only explained by missing datasets, it is also about how biological images are produced. Microscopy experiments are sensitive to experimental conditions, batch effects, and spatial or well and plate

---

[†]Equal contribution.

layout artifacts, and deep learning models can learn these signals instead of the underlying biology (Sypetkowski et al., 2023; Arevalo et al., 2024; Haslum et al., 2024). While these risks are widely recognized in principle, the field has not yet clearly demonstrated (using controlled, benchmark-focused analyses) how much such biases can distort model training and evaluation in microscopy. Importantly, a related biological modality has already made this issue concrete. In transcriptomics, several recent benchmarking studies have shown that confounding structure in the data can strongly influence performance estimates, sometimes to the point where simple statistical baselines rival or even outperform large foundation models (Luecken et al., 2022; Bendidi et al., 2024; Ahlmann-Eltze et al., 2025; Csendes et al., 2025). These results suggest that benchmark scores can reflect dataset-specific shortcuts rather than true biological generalization. Taken together, this motivates a careful look at whether microscopy benchmarks may face similar hidden failure modes, and whether current evaluations are reliably measuring what they intend to measure.

In this work, we investigate what microscopy foundation models truly learn by evaluating fondation models on curated cell-culture and tissue benchmarks, using ImageNet-1k as a natural-image reference. To make performance easier to interpret, we introduce two simple but informative baselines. First, *untrained models* use post-processed features from randomly initialized networks to test how much signal comes from architectural inductive biases and weak pixel correlations rather than learned biological content. Second, *disentangled tissue structures* represent histology images through the spatial organization of cells, testing whether models capture biological information beyond tissue morphology. We then benchmark widely used foundation models on these tasks and compare them to both baselines, finding that many models fall short of expectations.

## 2 RELATED WORKS

**Benchmarking pitfalls in biological machine learning.** As biological machine learning shifts toward large pretrained models, evaluation has become a central bottleneck: reported gains can reflect confounding structures (batch, lab, protocol, cohort, or platform effects) rather than a transferable biological signal. In microscopy, this issue is especially acute because experimental design and acquisition artifacts can imprint strong variation that models may exploit, motivating benchmark designs that explicitly test robustness to batch structure and technical covariates (Sypetkowski et al., 2023; Arevalo et al., 2024). Closely related concerns have been demonstrated in transcriptomics, where several benchmarking studies show that confounders and dataset shortcuts can inflate performance estimates, and that simple baselines can match or outperform large models on perturbation prediction (Bendidi et al., 2024; Ahlmann-Eltze et al., 2025; Csendes et al., 2025; Wenteler et al., 2025; Wenkel et al., 2025). Importantly, these results highlight why *strong simple baselines* (e.g., linear models, cell-count/viability proxies, or random-feature embeddings) are not an afterthought: they calibrate what a score means and help detect when progress comes from benchmark-specific shortcuts rather than biological abstraction (Ramanujan et al., 2020; Seal et al., 2025).

**Baselines for microscopy representations.** Across computer vision, untrained networks can already impose powerful image priors (Ulyanov et al., 2018; Heckel & Hand, 2019), and can exhibit non-trivial selectivity driven by architecture and initialization (Ramanujan et al., 2020; Baek et al., 2021; Kim et al., 2021). Related observations extend to transformer architectures, where substantial behavior can arise even when large parts of attention are fixed or random (Zhong & Andreas, 2024; Dong et al., 2025). In microscopy, simple and interpretable baselines such as CellProfiler-derived morphology features (Carpenter et al., 2006), cell-count / confluency summaries (Way et al., 2021; Seal et al., 2025), and spatial-organization representations that emphasize cellular arrangement in addition to cellular morphology (Wang et al., 2023) often provide competitive reference points. Including these baselines is essential to *calibrate* benchmark's difficulty and to relativize reported gains in presence of multiple sources of potential shortcuts. Training-light or training-free transfer pipelines for high-content screening further show that strong performance can arise from reusing generic representations with minimal task-specific training (Corbe et al., 2023).

**Microscopy imaging benchmarks.** Benchmarking in microscopy spans diverse assay families and thus benefits from separating *cellular-level* and *tissue-level* organizational scales. For cell culture, widely used public resources include curated collections (Masud et al., 2023) such as BBBC (Ljosa et al., 2012), perturbation imaging datasets designed to expose experimental batch effects

such as RxRx1 and related releases (Sypetkowski et al., 2023; Recursion, 2020), large-scale Cell Painting efforts such as JUMP-CP (Chandrasekaran et al., 2023), and more task-specific suites targeting heterogeneity across channels and acquisition settings (Chen et al., 2023). Recent work has also moved toward making large industrial-scale screens more usable for benchmarking by publishing compressed subsets and standardized tasks (Kraus et al., 2025; Sanchez et al., 2026). For tissue imaging, canonical benchmarks include challenge-style datasets such as CAMELYON16 and PANDA (Bejnordi et al., 2017; Bulten et al., 2022), alongside newer multimodal benchmarks that pair histology with molecular readouts (e.g., spatial transcriptomics) to evaluate cross-modal transfer (Jaume et al., 2024; Gindra et al., 2025; Bendidi et al., 2025). Complementarily, distribution-shift benchmark frameworks provide standardized splits and protocols for robustness testing across domains, including settings relevant to both pathology and microscopy (Koh et al., 2021).

**Microscopy imaging foundation models.** Microscopy foundation models are typically organized by modality. In cell culture imaging, large-scale self-supervised pretraining, using masked autoencoders and ViT-style backbones, has leveraged diverse Cell Painting datasets (Kraus et al., 2024; Watkinson et al., 2024; Kenyon-Dean et al., 2025), with extensions addressing channel and assay variability (Chen et al., 2023; Bourriez et al., 2024). For general-purpose fluorescence microscopy, models like Cytoself and SubCell aim to capture proteome-scale patterns (Kobayashi et al., 2022; Gupta et al., 2024). In histopathology, foundation models use both whole-slide and slide-level pretraining (Xu et al., 2024; Wang et al., 2024), self-supervised patch encoders tested across task suites (Chen et al., 2024), vision-language models (Lu et al., 2024), and multimodal approaches combining slides with reports and gene expression (Xu et al., 2025), with growing emphasis on scale and magnification diversity (Zimmermann et al., 2024; Bioptimus, 2025). Beyond pixel-based encoders, graph-based tissue models capture spatial cell organization and motivate structure-aware models (Wang et al., 2023). Given the dominance of ViT backbones, their architectural traits remain central to interpreting model behavior (Raghu et al., 2021; Darcet et al., 2023; Jiang et al., 2025).

## 3 BASELINES

Deep learning models can have performance confounders on microscopy benchmarks by exploiting low-level intensity cues or acquisition artifacts rather than biologically meaningful features. We therefore introduce a small set of intentionally simple baselines that (i) calibrate how well each model performs and (ii) help diagnose when an evaluation metric rewards shortcuts.

### 3.1 BASELINE STRATEGIES

We use three complementary baseline families to probe what information is sufficient to perform well: (1) *pixel-level statistics* that capture only global intensity distributions, (2) *untrained deep encoders* that isolate architectural inductive biases from learned biology, and (3) *disentangled tissue structure* representations that retain spatial cell organization while removing appearance and texture.

**Pixel-level baselines.** To test whether a task can be solved using low-level intensity correlations, we construct two feature sets from per-channel pixel statistics: `pixel_mean` (channel-wise means) and `pixel_stats` (channel-wise mean, standard deviation, and skewness). These features ignore spatial layout and morphology, and therefore serve as a simple lower bound that is easy to interpret.

**Untrained models.** Untrained models are deep networks with randomly initialized weights, used as fixed feature extractors with the same embedding pipeline as their trained counterparts. Because they contain no learned visual concepts, they separate the contribution of *architecture* (e.g., convolutional locality or tokenization in transformers) from the contribution of representation learning. When an untrained model performs competitively, it suggests that benchmark performance may be driven by low-level cues or dataset structure that aligns with architectural priors, and that the metric may not be faithfully reflecting biologically meaningful feature learning.

**Disentangled tissue structure.** In histology, the spatial organization of cells is biologically informative (e.g., it can reflect tissue types or pathological conditions like cancer) (Wang et al., 2023). At the same time, spatial structure can also become a shortcut: a model may succeed by recognizing

coarse patterns (cell density, layering, gland-like arrangements) without learning cell morphology or texture cues that are required for many clinically and biologically relevant distinctions. To study this explicitly, we introduce a *structure-only* baseline that retains cell positions while discarding pixel appearance, enabling us to measure how much of the downstream signal is explainable by the organization alone.

We formalize the notion of tissue structure as a set of 2D points defined by centroids of cell nuclei segmented by a pretrained segmentation model. In our experiments we use segmentation masks obtained from CellViT (Hörst et al., 2024) as provided by the authors of the benchmarks. We proceed to form two complementary views of this representation:

**(i) Binary images of cell graphs.** We build a cell graph from the centroids of segmented cell nuclei (e.g., using a simple spatial neighborhood rule) and render the resulting *edges* as binary images at the base resolution of the image encoders (224×224). This lets us apply both pretrained and untrained *image* models to structure alone, while disentangling completely the cell morphology and stain/texture. Because node (cell nuclei) locations are in a fixed coordinate system, the rendering is deterministic; we control apparent magnification and visibility by adjusting the drawing scale and edge width. We provide the construction details in Appendix B and include additional controlled studies on synthetic graphs in Appendix D.3. Fig. 10a provides an example of a tissue patch and its corresponding rendered cell-graph view.

**(ii) Point-cloud view of cell positions.** For completeness, we also treat the nuclei centroid set as a point cloud and apply geometric deep learning methods (Bronstein et al., 2021). This view naturally supports permutation invariance of nodes and makes it easier to test hypotheses about which aspects of organization matter (e.g., node count, density, local alignment, or mild position noise). While rotation invariance is desirable, some other common assumptions like symmetry with respect to mirror images and robustness to point density do not hold in general and are thus, not enforced. This representation also extends naturally to emerging 3D tissue settings (Lin et al., 2025).

## 3.2 EXPERIMENTAL SETUP.

To keep the analysis focused on representation quality rather than architectural novelty, we evaluated a set of widely used vision backbones and simple geometric encoders under a consistent embedding and evaluation pipeline. A detailed description of model configurations and pretrained weights is provided in Appendix C.

**Selected image architectures.** For image encoders we report both pretrained and randomly initialized variants, and probe intermediate layers to study how performance changes across depth. Concretely, we include: (i) a *single-layer CNN* (random local filters followed by global pooling) as a minimal inductive-bias baseline (`SingleConv`), (ii) *ResNet* models (He et al., 2016) pretrained on ImageNet-1k (Russakovsky et al., 2015) (and their random counterparts), and (iii) *ViT* models (Dosovitskiy et al., 2021) pretrained on ImageNet-21k (Ridnik et al., 2021) (and their random counterparts). Unless stated otherwise, we select four representative configurations per model family and evaluate four intermediate stages per network to compare early vs. late representations.

**Selected geometric architectures.** For cell-centroid point sets, we use standard message-passing GNNs in the MPNN framework (Gilmer et al., 2017), instantiating the neighborhood aggregation with *GCN* (Kipf & Welling, 2017) and *EGNN* (Satorras et al., 2021). Because there is no single widely adopted "structure-only foundation model" for these inputs, we complement the main benchmarks with controlled experiments on synthetic graphs to validate and interpret the behavior of these encoders (Appendix D.3).

## 3.3 BENCHMARKING SETUP

We evaluate representations in two organizational scales for microscopy imaging: the cellular and tissue levels, chosen to probe complementary biological scales. These scales are represented by images of cell culture and WSIs of extracted tissue samples. We include a natural-image reference to highlight where common representation learning intuitions do and do not transfer. Across all benchmarks, we extract frozen embeddings and follow the dataset-specific aggregation and evaluation protocols described below (please refer to Appendix B for modality details).

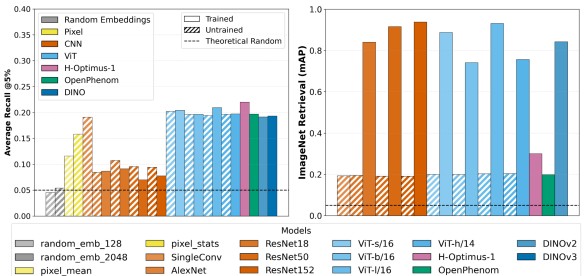 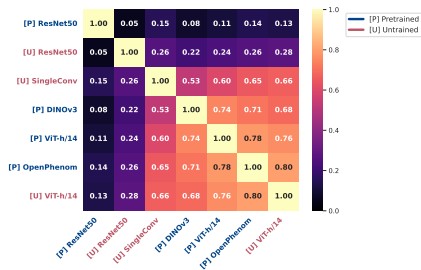

(a) Comparison between trained and untrained models on natural images and Cell Painting data from RxRx3-core. *Right*: kNN top-1 accuracy on ImageNet-1k across models. *Left*: gene–gene retrieval performance on RxRx3-core (Recall@5%). Mean recall across all 5 literature datasets.

(b) Spearman correlations between gene–gene similarity rankings induced by different representations on RxRx3-core genes. Models are hierarchically clustered to reveal functional groupings.

Figure 1: **Comparison between trained and untrained models.** (a) Task-dependent metrics on natural images and Cell Painting data. (b) Correlations between predictions on Cell Painting data. Large scale subfigures are available in I.

**Cell culture.** We evaluate Cell Painting representations using two retrieval benchmarks. On RxRx3-core, we perform gene–gene retrieval following Kraus et al. (2025), measuring whether genes with shared biological function are ranked among the most similar representations. On JUMP-CP, we assess chemical perturbation retrieval using the pipeline of Sanchez et al. (2026), reporting five-fold mAP on shared positive controls. Full dataset construction, metric definitions, and fold protocols are detailed in Appendix. B.2.

**Tissue.** To probe representations and analyze structure-aware baselines, we use spatial transcriptomics (ST) samples from *HEST-1k* (Jaume et al., 2024), which pairs H&E patches with gene expressions. This setting is particularly diagnostic: accurate prediction across many genes requires embeddings that integrate information spanning single-cell morphology and the surrounding tissue context. We follow the original benchmarking protocol, however, to better understand the contribution of structure-only inputs we additionally implement a coarser version of the benchmark. It consists of *binned* neighborhood patches and we refer to as HEST-1k-1NN. A detailed comparison between the two and the corresponding experimental results are provided in Appendices E and F.

**Natural images.** To contextualize microscopy results, we include a standard natural-image reference using *k*NN probing on ImageNet-1k. This comparison highlights how representation quality evolves across intermediate layers in a setting where high-level semantic features are known to emerge reliably, and clarifies which observations in microscopy are genuinely atypical rather than artifacts of our evaluation pipeline.

## 4 RESULTS

### 4.1 CELLULAR LEVEL

Fig. 1a shows gene-gene retrieval performance on RxRx3-core (recall@5% of cosine similarities). Unexpectedly, untrained ViTs perform comparably to pretrained ViTs and foundation models like OpenPhenom. Moreover, a minimal untrained SingleConv baseline is competitive with the best-performing methods, while pixel-statistics baselines recover a substantial fraction of the signal. In contrast, ResNet representations, whether pretrained or untrained, perform poorly on this task.

The results are stable across three random seeds and three folds, indicating that weight initialization does not impact untrained models (table 10 for RxRx3-Core and table 11 for JUMP-CP).

Following Zhong & Andreas (2024), who attribute the success of untrained models on certain tasks to their ability to identify suitable low-dimensional subspaces, we analyze the relationship between representation dimensionality and benchmark scores. PCA reveals that the best performing models occupy similar high-dimensional spaces achieving similar relationship recall (Fig. 16).

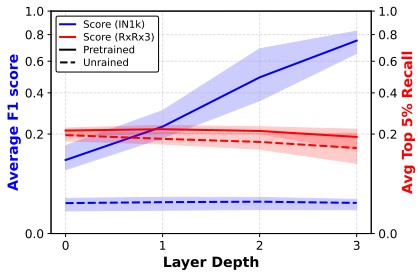

(a) RxRx3-core, gene-gene relationship retrieval.

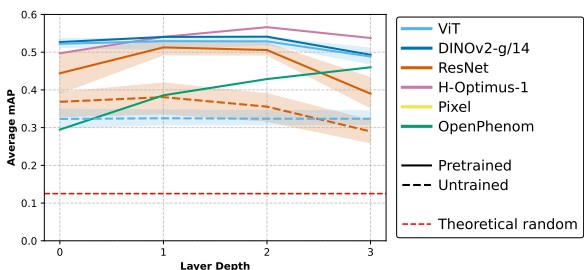

(b) JUMP-CP, replicate retrieval.

Figure 2: **Comparison of performance of intermediate layers.** Minimal/ maximal scores are reported. (a) Classification on ImageNet-1k (in blue) strongly favors deeper layers of pretrained models unlike the relationship retrieval tasks on RxRx3-core (in red). (b) Replicate retrieval for a subset of JUMP-CP compounds grouped by architecture.

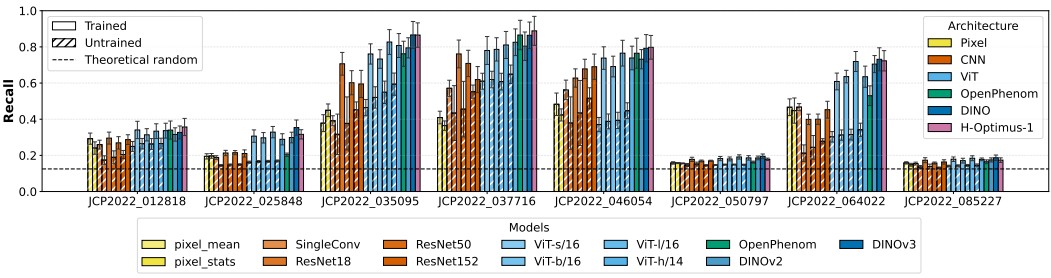

Figure 3: **Mean average precision (mAP) per compound on the JUMP-CP benchmark.** Bar plots show mAP scores for eight positive-control compounds and the mean across compounds. Each bar corresponds to a model configuration, grouped and colored by architecture family. Solid bars denote pretrained models, hatched bars denote untrained models, and pixel-based baselines are included for reference. Error bars indicate variability across evaluation folds. An upscaled version is available in I.

To validate whether models retrieve the *same* gene relationships, we compute gene-gene similarity rankings across all genes (270k pairs) for each model and measure pairwise Spearman correlations between rankings (Fig. 1b for a selected subset of models and Fig. 21 for all models). Pretrained and untrained ViTs yield highly correlated rankings, indicating shared relational structure; their rankings also correlate with DINOv3 and the `SingleConv` baseline.

Finally, for each model, we select four intermediate layers to evaluate. By averaging performance across architectures and layers (Fig. 2), we observe that performance generally decreases or remains stable in deeper layers for biological recall, in contrast to the monotonic improvements typically observed on natural image benchmarks (Fig. 2a). Here, all models except for the untrained ones demonstrate an increase in F1 score when evaluating hidden representations from deeper layers.

For the JUMP-CP benchmark, per-compound mean average precision (mAP) results are shown in Fig. 3. Several compounds (e.g., JCP_050797 and JCP_085227) exhibit near-random retrieval performance across all models, suggesting little to no detectable phenotypic effect. Other compounds (e.g., JCP_012818) are retrieved almost equally well by untrained baselines and pretrained models, indicating easily detectable phenotypic changes. In contrast, some compounds (e.g., JCP2022_025848) are only reliably retrieved by pretrained or foundation models, suggesting that these models capture more subtle discriminative features. Overall, pretrained and foundation models consistently outperform untrained and pixel-based baselines. However, untrained baselines provide an important reference point for quantifying the amount of signal gained through learning. For example, although JCP2022_037716 achieves a higher absolute score than JCP_046054, the relative improvement over untrained baselines is larger for JCP_046054.

Another layer-wise effect emerges on JUMP-CP (Fig. 3). For untrained baselines and out-of-domain (OOD; e.g., ImageNet-pretrained backbones) models, performance typically saturates or slightly decreases in the final layers. In contrast, in-domain (IID) models trained on similar cell-painting data (e.g., OpenPhenom) tend to improve with depth, with the largest gains observed for a subset of compounds (Fig. 23).

To complete this part of our analysis with a cautiously optimistic perspective we evaluate the recently released RxRx3-core embeddings from MAE-L/8 (Kraus et al., 2024) and MAE-G/8 (Kenyon-Dean et al., 2025) models. These backbones were pretrained on large proprietary in-distribution datasets as described in (Kraus et al., 2024) and (Kenyon-Dean et al., 2025) respectively. Benchmarking on a balanced split of RxRx3-core shows a considerable improvement over all pretrained and untrained baselines, reaching 1.5 times higher averaged top 5% recall (Appendix H). Since the pretrained backbones remain private we cannot proceed with our layer-wise comparative setup or study the generalization to other datasets. While this limits our conclusions about the acquisition of relevant high-level abstractions, the obtained results are an encouraging sign for further investigations.

## 4.2 TISSUE LEVEL

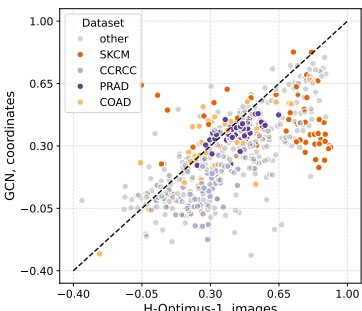 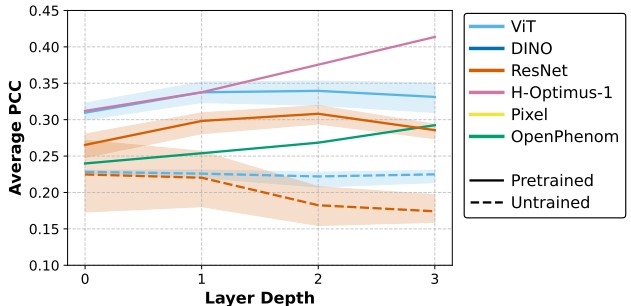

(a) Gene-wise average test PCC for each of the dataset specific regression targets across cross-validation folds.

(b) Global average PCC for OOD vision backbones and foundation models trained on bioimaging modalities. Minimal/ maximal scores across model configurations are reported.

Figure 4: **Analysis of results on HEST-1k-1NN.** (a) Comparison between the best structure-only model and H-Optimus-1. (b) Performance analysis of intermediate layers.

In table 1 we present averaged metrics for the *best performing model* for each selected dataset-modality-training setting (scores of individual models can be found in Appendix F). As expected, in-domain foundation models offer a substantial increase in performance across most datasets. Surprisingly, however, for COAD and PRAD the performance of structure-based models is competitive with OOD vision encoders.

To further investigate this phenomenon we compare performance of the best image model against the best structure model for each of the 50 target genes in Fig. 4a. On PRAD and COAD, we observe a surprisingly competitive scores between a structure-only encoder and H-Optimus-1 for a large subset of genes. The foundation model achieves an impressive PCC of >0.8 on several genes, while struggling to reach positive correlation with several genes well predicted from cell coordinates. Our structure-based baseline reaches its best scores on SKCM suggesting further questions regarding relevant biological functions of the associated genes.

Table 1: **Performance of the best-in-class model on a selection of HEST-1k-1NN datasets.** Mean and standard deviation across cross-validation folds are reported. The best structure-based model were pretrained on natural images.

| MODALITY | TRAINING | CCRCC | PRAD | SKCM | COAD |
|---|---|---|---|---|---|
| FULL IMAGE | NO | 0.12 ± 0.14 | 0.25 ± 0.06 | 0.34 ± 0.06 | 0.27 ± 0.08 |
| FULL IMAGE | YES, OOD | 0.26 ± 0.07 | 0.41 ± 0.01 | 0.62 ± 0.07 | 0.30 ± 0.07 |
| FULL IMAGE | YES, IID | **0.37** ± 0.06 | **0.49** ± 0.02 | **0.68** ± 0.03 | **0.38** ± 0.06 |
| STRUCTURE | YES, OOD | 0.16 ± 0.04 | 0.38 ± 0.04 | 0.45 ± 0.07 | 0.31 ± 0.01 |

We then compare predictions of pretrained and untrained models. Table 1 shows untrained models to offer better-than-random performance and help provide additional context about the difficulty of the target. Notably, the performance of random encoders on SKCM is more than 2.5 times higher than on CCRCC, despite the task remaining the same.

Nonetheless, in contrast to the experimental results on RxRx3-core, subsets of HEST-1k like CCRCC and COAD demonstrate that in-domain pretraining offers a clear advantage. We proceed to investigate whether pretrained models form relevant deep-layer features (Fig. 4b). Indeed, the performance of intermediate layers improves with depth for both IID and OOD models. Moreover, for the in-domain H-Optimus-1 the outputs of the very last layer provide the best results, suggesting that the model has learned relevant high-level features. Interestingly, OpenPhenom, pretrained on images of cell culture demonstrates a similar trend. Additionally, H-Optimus-1 shows strong performance in experiments on OOD JUMP-CP and RxRx3-core (Fig. 3 and Fig. 15b respectively).

Finally, in the absence of established foundation models for structure-only tissue representations we investigate different pre-training strategies and compare them with random initialization. Our results show that pretraining on synthetic data and natural images favors some architectures. We provide details in Appendix E.4.

## 5 DISCUSSION

### 5.1 WHAT CURRENT MICROSCOPY BENCHMARKS CERTIFY

Across cell culture (RxRx3-core, JUMP-CP) and tissue (HEST-1k, HEST-1k-1NN), the results show that benchmark performance does not consistently track acquisition of high-level biological abstractions. On RxRx3-core, several pretrained and foundation models perform comparably to intentionally simple baselines, implying that part of the necessary signal is accessible without learning biologically meaningful representations. In contrast, on selected subsets of JUMP-CP and HEST-1k, in-domain pretraining yields clearer gains, indicating that learning can matter when the task exposes subtler or modality-aligned signals. Overall, the experiments support a conservative reading of absolute scores: they can reflect a mixture of biology, architectural priors, and low-level correlates rather than biological abstractions alone.

### 5.2 BASELINES AND DIAGNOSTICS THAT CHANGE INTERPRETATION

Therefore, strong simple baselines are necessary to interpret a score. On RxRx3-core, untrained ViTs and `SingleConv` are competitive with pretrained models, while pixel-statistics features achieve non-trivial recall despite discarding spatial structure. Moreover, pretrained and untrained ViTs exhibit highly correlated gene–gene similarity rankings (Spearman), implying that they exploit essentially the same relational signal under this benchmark. These observations are direct evidence that architectural inductive bias and dataset-accessible cues can dominate measured performance on this task. Separately, representational collapse (high variance explained by the top PCA components) coincides with weak biological recall for ResNet-based representations, while higher-dimensional embeddings (ViTs, pixel baselines) align with stronger recall; the same dimensionality diagnostic separates successful vs. unsuccessful behavior in ImageNet-1k $k$NN probing. Effective dimensionality is therefore a useful sanity check for collapse, but it does not establish biological semantics.

### 5.3 SPATIAL ORGANIZATION IN TISSUE IS A STRONG MODALITY, WITH OPEN CAUSAL QUESTIONS

On HEST-1k-1NN, structure-only models (cell-centroid graphs) approach and sometimes exceed OOD image baselines on selected tissue categories (notably PRAD and COAD), and gene-wise analyses show subsets of genes that are comparatively predictable from cell coordinates. At the same time, in-domain histology foundation models remain the best on several subsets (e.g., CCRCC, SKCM), demonstrating that morphological cues learned from histology provide additional information beyond organization alone. This suggests further exploration of methods that integrate this signal in a disentangled interpretable way. A plausible alternative explanation for structure-only performance is correlation with local cell count: the cell-counting baseline is strong, and while structure-based models can demonstrably capture additional organizational features (as studied on

synthetic data) it does not yet prove that arrangement (rather than count) drives the gains on real tissue. Stronger evidence would require matching analysis (conditioned on cell count), or controlled interventions beyond synthetic data that vary arrangement while holding count fixed.

## 6  FUTURE WORK AND LIMITATIONS

The scope of the presented work focuses on re-contextualizing benchmark results by introducing baselines that are sensitive to simple shortcuts. However, to see if the models leverage these shortcuts, we rely primarily on prediction alignment, inviting further interpretability work. Another interesting direction is to explore the formation of high-level abstractions through the lens of representation alignment across in-domain, out-of-domain, and baseline models at the global and local scales (Huh et al., 2024), (Gröger et al., 2026).

Additionally, the impact of structure as a standalone modality suggests revisiting graph-based representations for tissue imaging (Wang et al., 2023), (Pati et al., 2022), (Öğüt et al., 2025). Notably, a promising direction would be to explore the disentangled fusion of structural information with local cell morphology as well as the models' susceptibility to structural biases (e.g. cell count), to develop more robust pretraining and evaluation tasks.

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

# A  APPENDIX

## A  THE USE OF LARGE LANGUAGE MODELS (LLMS)

LLMs were used for the sole purpose of refinement of written text (i.e. proofreading and paraphrasing for the clarity of presentation).

## B  DATA MODALITIES AND DATASETS

### B.1  NATURAL IMAGES

**ImageNet-1k.** As a well-studied reference dataset of natural images we use ImageNet-1k (ILSVRC 2012) as described in (Russakovsky et al., 2015). The dataset consists of 1000 classes and is split into train, validation and test subsets (of 1,281,167 / 50,000 / 100,000 images respectively). In our experiments we use the train / validation split. Samples from the 8-class subset used for mAP computation (Fig. 19b) can be seen in Fig. 5. The classes were chosen based on KNN performance of DINOv2-g/14, more specifically we take the two best and the two worst classes with respect to their validation F1 scores. The remaining four classes were sampled randomly.

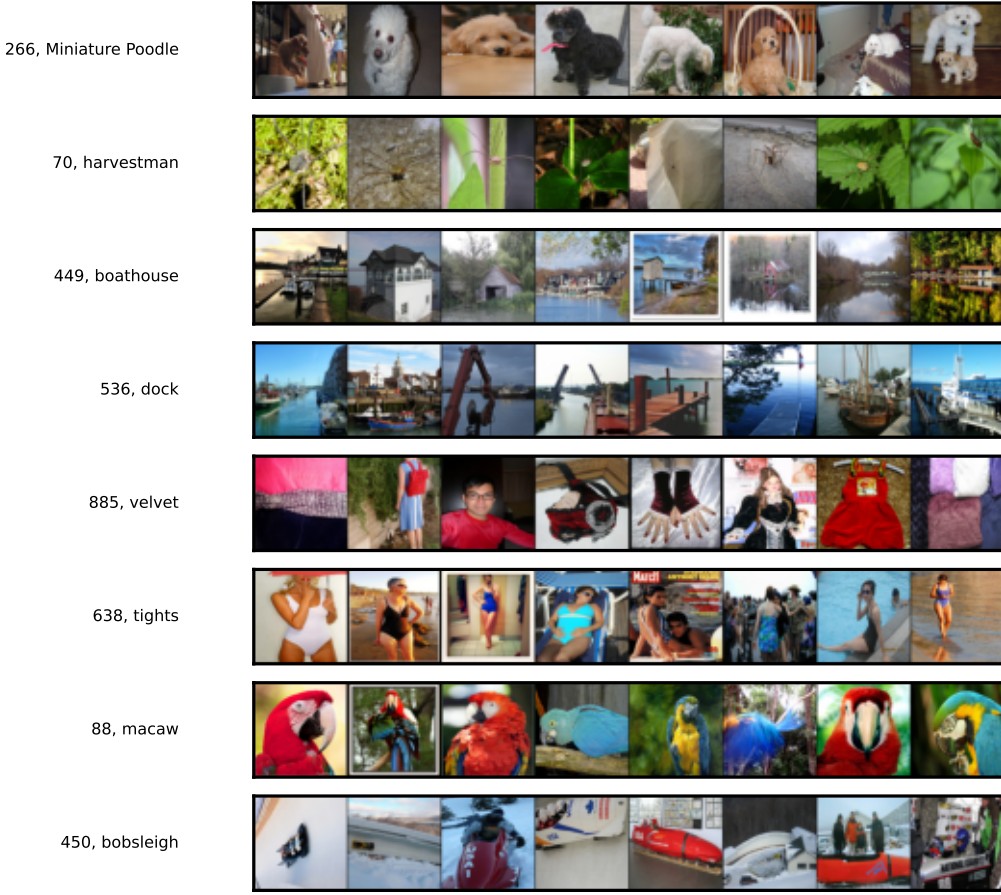

266, Miniature Poodle

70, harvestman

449, boathouse

536, dock

885, velvet

638, tights

88, macaw

450, bobsleigh

Figure 5: Samples from 8 classes sampled for the mAP experiment.

B.2  CELL PAINTING

Cell Painting (Bray et al., 2016) is a standardized fluorescence microscopy assay that captures diverse visual characteristics of cells at single-cell resolution. Cells are labeled using a fixed set of fluorescent markers, each highlighting a different cellular component (compartments and organelles, such as the nucleus, cytoskeleton, mitochondria, endoplasmic reticulum, and plasma membrane), and imaged across multiple channels to produce rich, multi-channel microscopy images. Images are typically acquired across five to six fluorescence channels, producing multi-channel microscopy images that encode diverse aspects of cell morphology, organization, and subcellular structure.

Cell Painting is widely used in large-scale biological and pharmaceutical studies, including drug discovery, genetic perturbation screening, and functional genomics. Its central premise is that perturbations that affect similar biological pathways induce similar cellular phenotypes, which can be detected through changes in cell morphology and organization. As a result, Cell Painting has become a key modality for representation learning, where the goal is to learn feature embeddings that capture biologically meaningful variation across perturbations with or without task-specific supervision.

From a machine learning perspective, Cell Painting datasets pose several challenges. They are high-dimensional, multi-channel, and exhibit strong sources of variation unrelated to biological signal, such as technical, called batch effect, imaging artifacts, and cell-cycle heterogeneity. Moreover, biological semantics are indirect: labels often correspond to treatments or genes rather than explicit visual concepts. Consequently, evaluating representation quality is non-trivial and typically relies on downstream proxy tasks, such as perturbation matching or gene–compound association retrieval.

These properties make Cell Painting pertinent for understanding whether representation learning methods can move beyond appearance-driven features and capture higher-level, biologically meaningful abstractions.

**JUMP-CP**  JUMP-Cell Painting is a large-scale microscopy dataset generated by the Joint Undertaking for Morphological Profiling (JUMP) Consortium, a collaboration between ten pharmaceutical companies, six technology partners, and two non-profit organizations (Chandrasekaran et al., 2023). The dataset comprises Cell Painting images of human osteosarcoma (U2OS) cells subjected to diverse perturbations, including chemical treatments, gene overexpression, and CRISPR-Cas9 knockouts. JUMP-CP includes over 116,750 compounds, 12,602 gene overexpression perturbations, and 7,975 gene knockouts, totaling approximately 115 TB of data and capturing single-cell profiles for more than 1.6 billion cells. Each experimental compound plate, across all batches and laboratories, contains the same eight positive-control compounds (Fig. 6) and negative controls (DMSO only). We used these shared controls are used to define a standardized benchmark for evaluating representation robustness across experimental conditions.

**Our JUMP-CP Subset Benchmark**  We construct a balanced evaluation benchmark by selecting five folds, each composed of four experimental compound plates from each the seven laboratories providing 384-well plates. Across these $28 \times 5$ plates, we use all control wells, including positive controls for retrieval metric computation and negative controls (DMSO) for batch-effect mitigation. This results in approximately $6,400 \times 5$ five-channels images, forming a diverse subset in terms of both laboratory origin and phenotypic variation, which we use to evaluate produced representation of diverse encoders.

**RxRx3-core.**  RxRx3 is a large-scale microscopy benchmark designed to evaluate representation learning for cellular imaging under biologically relevant perturbation tasks. It comprises fluorescence six-channels Cell Painting images of cells subjected to both gene knockdown and compound perturbations across multiple experimental batches. The RxRx3-core subset (Kraus et al., 2025) is a publicly available benchmark that defines gene-gene and gene-compound retrieval tasks, in which representations are evaluated based on their ability to retrieve matching perturbations in an unsupervised setting, despite substantial biological and experimental variability. The initial subset from Kraus et al. (2025) contains 222,601 microscopy images spanning 736 CRISPR knockouts and 1,674 compounds at 8 concentrations. Yet, the dataset exhibits substantial variability in the number of image replicates per gene (ranging from ~50 to ~9,450), along with opportunities for improved quality control (Fig. 7). Therefore, we adopt a subsampled three-fold version of the dataset, using 10 images per gene.

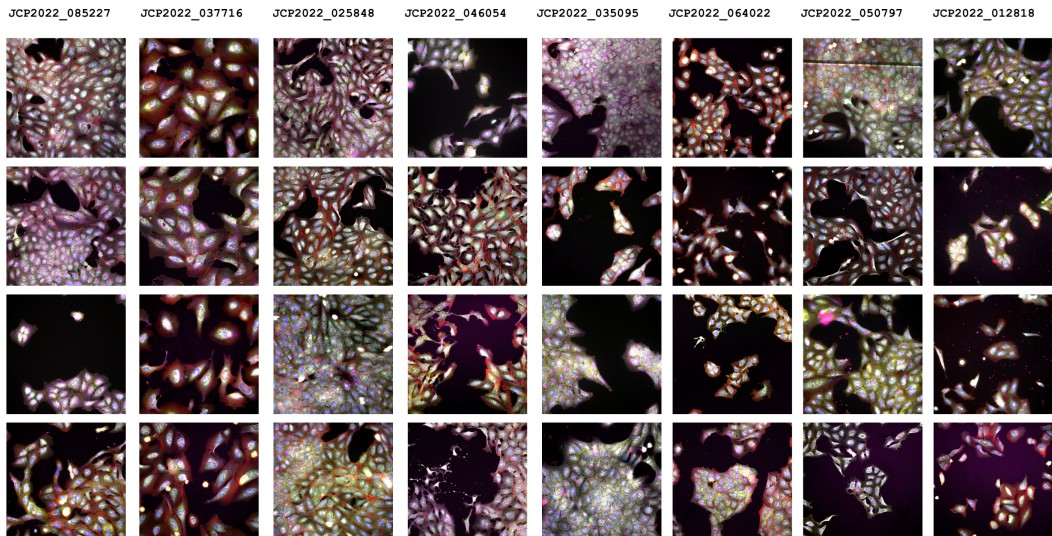

Figure 6: **Example of images from the 8 positive controls of JUMP-CP.¡**

**Retrieval tasks and metrics.** For both benchmarks, representations are evaluated through retrieval-based proxy tasks that assess whether biologically related perturbations are embedded nearby in representation space.

**JUMP-CP (compound retrieval).** Following Sanchez et al. (2026), compound profiles are constructed by aggregating image features across control wells. Each compound profile is treated as a query and ranked against all others using cosine similarity. Performance is quantified using mean average precision (mAP), which measures whether replicate profiles of the same compound are retrieved ahead of profiles from other compounds. Evaluation is performed using five folds.

**RxRx3-core (gene–gene retrieval).** Following Kraus et al. (2025), gene representations are obtained by aggregating image-level features across replicates. Each gene is used as a query to rank all other genes by cosine similarity. Performance is measured as recall of literature-supported functional gene pairs within the top 5% most similar ranks. We restrict evaluation to the upper similarity tail, as the lower tail contains negligible biological signal (Fig. 20). We use three folds to enable variance estimation.

Unless otherwise stated, results are reported as mean ± standard deviation across folds.

### B.3 TISSUE IMAGING AND SPATIAL TRANSCRIPTOMICS

**Whole Slide Images (WSI)** are microscopy images of thin slices of tissue. To facilitate analysis of tissue morphology the slides are often stained with hematoxilyn and eosin (H&E staining). Hematoxilyn reveals cell nuclei in a darker purple-blue color against the cytoplasm colored pink by eosin. Typically WSIs are large in size and resolution making them prohibitively expensive to process as single images. Thus, dividing them into patches possibly at different scales becomes necessary (Fig. 10a). Different scales exhibit varying levels of feature hierarchy. For instance in context of cancer detection and sub-typing high resolution local patches can reveal morphological differences from the normal cells, while more global views allow for instance to assess the extent of tumor proliferation in affected tissues.

**HEST-1k** is a dataset of 1,229 spatial transcriptomic samples as introduced in Jaume et al. (2024) covering 26 organs from two different species. The full dataset consists of 2.1 million image patches matched to gene expression profiles from ST spots. Full experimental details can be found in Jaume et al. (2024). We focus only on a subset of data. Namely, a subset of H&E-stained slides is used for the *HEST-1k benchmark*. All image samples consist of WSI crops at 112×112 $\mu$m which corresponds to 20× magnification at 224×224 pixel resolution. The estimated pixel size for stan-

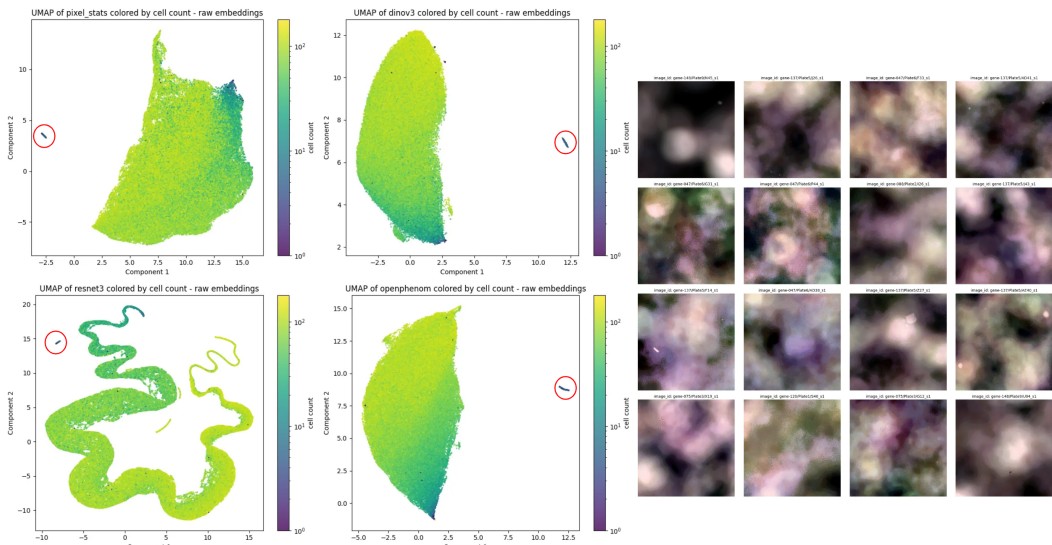

Figure 7: **Outlier image detection on RxRx3-Core using raw image representations. Left:** UMAP projections of embeddings produced by different encoders colored by cell count, highlighting an outlier cluster in red. **Right:** Representative images sampled from the identified outlier cluster.

Table 2: Benchmark subsets of HEST-1k as defined in Jaume et al. (2024). We keep the original structure of the benchmark unchanged.

| Task | Number of slides | Number of patients (splits) | Condition |
|---|---|---|---|
| IDC | 4 | 4 | Invasive ductal carcinoma (breast) |
| PRAD | 23 | 2 | Prostate adenocarcinoma |
| PAAD | 2 | 2 | Pancreatic adenocarcinoma |
| SCKM | 2 | 2 | skin cutaneous melanoma |
| COAD | 4 | 2 | colon adenocarcinoma |
| READ | 4 | 2 | rectal adenocarcinoma |
| ccRCC | 24 | 24 (6) | clear cell renal carcinoma |
| LUAD/LUNG | 2 | 2 | lung adenocarcinoma |
| LYMPH IDC | 4 | 2 | see IDC |

dardizing resolution between slides is provided with the dataset, allowing us to scale the relative coordinates of the detected cells. For each image as set of 50 genes with the highest normalized variance is provided. The benchmark implements an evaluation pipeline consisting of PCA-reduction of pre-extracted patch embeddings resulting into 256-dimensional vectors which are then used by a trainable linear ridge regression head on the 50 gene expression targets. The gene-wise Pearson correlation coefficients (PCC) are computed for each cross validation fold and are averaged across each subset of the benchmark before being averaged into a global score. We follow the evaluation protocol provided by the authors via https://github.com/mahmoodlab/HEST.

We detail the subsets and the number of corresponding samples in table 2.

**HEST-1k-1NN** is a coarsened version of HEST-1k benchmark dataset obtained by binning of adjacent spatial transcriptomics spot. More precisely, for each patch from the original dataset we extract the absolute coordinates of the corresponding ST spot which can be given by the coordinates of the corner of the patch or by its centroid. The coordinates of the spots form a square or hexagonal grid depending on the benchmark and for each patch we aggregate up to 9 and 7 total spots respectively. Incomplete neighborhoods at the edges of a slide are not discarded, instead we drop empty patches with no detected cells. The neighbor count distribution is given in Fig. 8. The number of samples in the coarsened dataset has the same order of magnitude as the original one.

Following the coarsening of the ST grid, for each aggregated neighborhood of a spot a bounding box of all constituent image patches is computed. Then, the original WSI is cropped at the location given by the bounding box accounting for the adjustments in resolution as provided in the benchmark metadata.

This strategy yields a dataset of aggregated overlapping image patches. Both the source and the resulting aggregated patches overlap, with the former being caused by the patches being larger than the physical ST spots. Importantly, we do not introduce any patch-based splits. The absence of data leakage is ensured by following the patient-stratified train-test splits as described by the authors of the benchmark (Jaume et al., 2024).

Finally, the target counts are averaged across patches additionally providing a type of smoothing over noisy ST readouts.

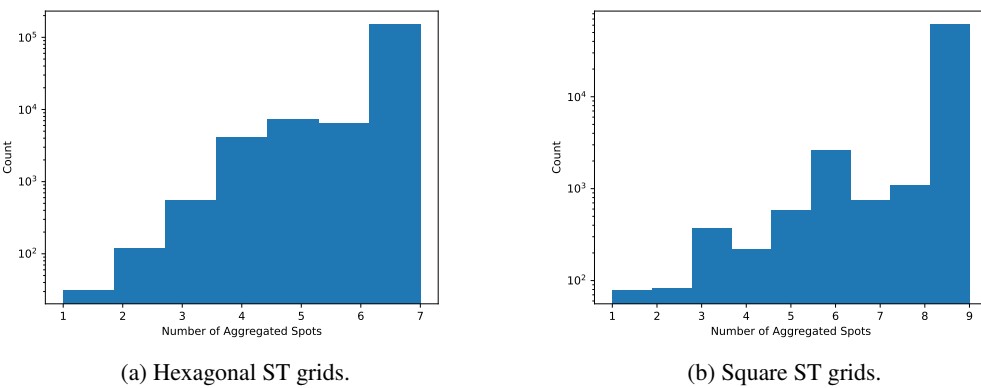

(a) Hexagonal ST grids.  (b) Square ST grids.

Figure 8: We display neighbor counts for slides with different ST grids. The counts are given in the logarithmic scale.

## C  BASELINE MODELS

### C.1  OUT OF DOMAIN MODELS

Table 3: Configurations of ResNet He et al. (2016) and ViT Dosovitskiy et al. (2021) used in our experiments.

| Architecture | `timm` Model Name | `timm` Weights | Model Stages | Aggregation |
|---|---|---|---|---|
| *Convolutional Neural Networks (CNNs)* | | | | |
| ResNet18 | `resnet18` | `tv_in1k` | layer $\in \{1, 2, 3, 4\}$ | Avg-pool |
| ResNet34 | `resnet34` | `tv_in1k` | layer $\in \{1, 2, 3, 4\}$ | Avg-pool |
| ResNet50 | `resnet50` | `tv2_in1k` | layer $\in \{1, 2, 3, 4\}$ | Avg-pool |
| ResNet152 | `resnet152` | `tv2_in1k` | layer $\in \{1, 2, 3, 4\}$ | Avg-pool |
| *Vision Transformers (ViTs)* | | | | |
| ViT-s/16 | `vit_small_patch16_224` | `augreg_in21k_ft_in1k` | block $\in \{3, 6, 9, 12\}$ | `<cls>`-token |
| ViT-b/16 | `vit_base_patch16_224` | `augreg2_in21k_ft_in1k` | block $\in \{3, 6, 9, 12\}$ | `<cls>`-token |
| ViT-l/16 | `vit_large_patch16_224` | `augreg_in21k_ft_in1k` | block $\in \{6, 12, 18, 24\}$ | `<cls>`-token |
| ViT-h/14 | `vit_huge_patch14_224` | `orig_in21k` | block $\in \{8, 16, 24, 32\}$ | `<cls>`-token |

**AlexNet**  is a convolutional neural network architecture originally developed for large-scale image classification. We evaluate the standard AlexNet architecture using both ImageNet-pretrained weights and randomly initialized (untrained) weights, as provided by the `torchvision` (Wightman, 2019) library.

**ResNets and Vision Transformers**  are listed in table 3. All models and intermediate stages are is integrated into the HEST-1k benchmarking suite and evaluated using the provided protocol.

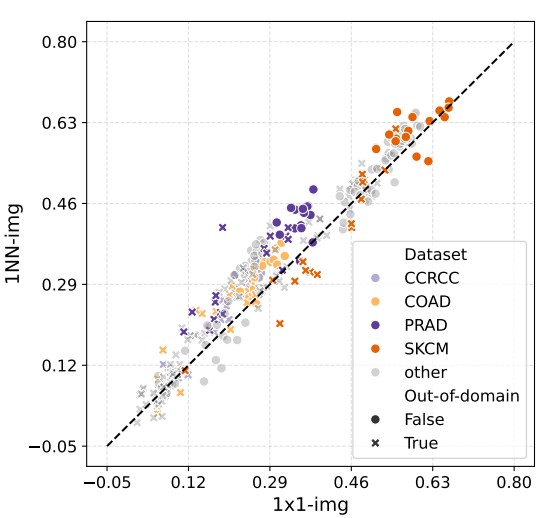 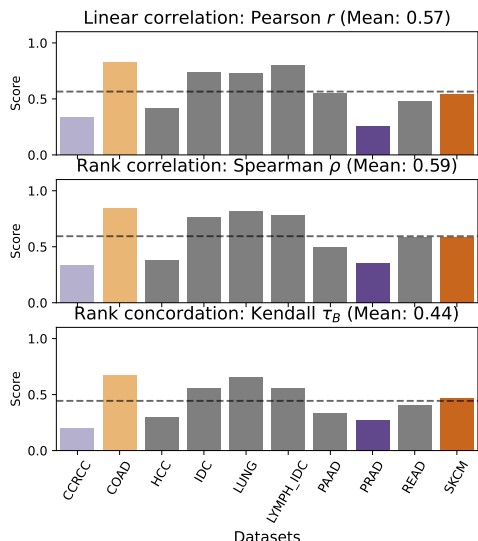

(a) Alignment between the performance of models on HEST-1k and HEST-1k-1NN. Each point represents dataset-wise PCC for a given model configuration. The plot includes in domain and out of domain models.

(b) Correlations between the performance of models on HEST-1k and HEST-1k-1NN. The correlations are computed for *in domain foundation models* only

Figure 9: Comparison between HEST-1k and HEST-1k-1NN. (a) Alignment of average PCC for a given dataset-model configuration pair. (b) Correlations between the performance of histology foundation models.

**DINOv2 & DINOv3** are self-supervised vision transformers trained using teacher–student distillation, with DINOv3 further incorporates larger-scale training. We evaluated the `dinov2_vitg14` and `dinov3_vit7b16` architectures using pretrained weights released through the official `facebookresearch` DINOv2 and DINOv3 repositories

## C.2 In Domain Models

**OpenPhenom** is a masked auto-encoder foundation model based on a ViT-s/16 architecture, pretrained directly on large-scale cell culture microscopy data. We evaluate OpenPhenom using the publicly released pretrained weights available from the Hugging Face repository `recursionpharma/OpenPhenom`. We discard the `<cls>`-tokens and use the average of patch tokens following the implementation from `https://huggingface.co/recursionpharma/OpenPhenom`. For stage-wise experiments we use blocks {3, 6, 9, 12}. The model is integrated into the HEST-1k benchmarking suite and evaluated using the provided protocol.

**MAE-L/8, MAE-G/8** are masked auto-encoder foundation models, pretrained on large-scale (93 and 16 million images respectively) cell culture microscopy datasets as described in (Kraus et al., 2024) and (Kenyon-Dean et al., 2025). The pretrained backbones are not publicly available, however the embeddings of RxRx3-core can be accessed via the official Hugging Face repository `https://huggingface.co/datasets/recursionpharma/rxrx3-core`.

**UNI / UNI 2** as introduced in Chen et al. (2024) are histology (designated for tissue analysis) foundation models based on ViT-l/14/ ViT-h/14-reg8 architectures respectively. The models were trained on patches from 100,000 / 350,000 histology slides using DINOv2 (Oquab et al., 2024). The models are evaluated using the HEST-1k benchmarking suite.

**CONCH (v1, v1.5)**  Lu et al. (2024) are ViT-b and ViT-l based histology foundation models, fine-tuned from UNI checkpoints with iBOT (Zhou et al., 2022) on 1.17 million of histology image/ caption pairs. The models are evaluated using the HEST-1k benchmarking suite.

**Kaiko Base 8**  as introduced in kaiko.ai et al. (2024) is a histology foundation model based on the ViT-b/8 architecture, trained with DINO (Caron et al., 2021). The model is evaluated using the HEST-1k benchmarking suite.

**GigaPath**  as introduced in Xu et al. (2024) is a histology foundation model based on the ViT-g architecture, trained with DINOv2 on patches from 171,189 WSIs. The model is evaluated using the HEST-1k benchmarking suite.

**Hibou Large**  as introduced in Nechaev et al. (2024) is a histology foundation model based on the ViT-l architecture, trained with DINOv2 on patches from 1,138,905 WSIs. The model is evaluated using the HEST-1k benchmarking suite.

**Phikon (v1 / v2)**  as introduced in Filiot et al. (2023) / Filiot et al. (2024) are histology foundation models based on ViT-b / ViT-l trained with iBOT / DINOv2 on patches from 6,093 / 60,000 WSIs. The model is evaluated using the HEST-1k benchmarking suite.

**CTransPath**  is a histology foundation model based on a Swin Transformer (Swin-T/14) (Liu et al., 2021) architecture and trained using MoCoV3 on patches from 32,220 WSIs. The model is evaluated using the HEST-1k benchmarking suite.

**Virchow / Virchow2**  as introduced in Vorontsov et al. (2024) and (Zimmermann et al., 2024) respectively are histology foundation models based on the ViT-h architecture trained on patches from 1.5 million / 3.1 million WSIs using DINOv2. The model is evaluated using the HEST-1k benchmarking suite.

**H-Optimus-0**  as introduced in Saillard et al. (2024) is a histology foundation model trained with DINOv2 on patches from 500,000 WSIs. The architecture is based on a 40-block ViT-g/14. The model is evaluated using the HEST-1k benchmark. In our stage-wise experiments we use `<cls>`-tokens of blocks $\{10, 20, 30, 40\}$.

**H-Optimus-1**  is a histology foundation model trained with DINOv2 on patches from more than 1 million slides (Bioptimus, 2025). The architecture is based on a 40-block ViT-g/14. The model is evaluated using the HEST-1k benchmarking suite. In our stage-wise experiments we use `<cls>`-tokens of blocks $\{10, 20, 30, 40\}$.

## C.3  Graph Neural Networks

In absence of commonly used foundation models for *structure-only* cell graphs and point sets we restrict ourselves to simple GNN baselines, demonstrating their potential in controlled settings, and leaving architectural exploration for future work.

**Graph Convolutional Network (GCN)**  is an architecture introduced in Kipf & Welling (2017) and defined by a specific aggregation of graph neighborhoods, which was inspired by a first-order approximation of graph spectral convolutions. It can be viewed as an instance of message passing and extension of CNNs to irregular grids (Gilmer et al., 2017; Bronstein et al., 2021).

We define our GCN-based model as a 4-layer convolutional network operating on node embeddings given by a learned linear projection of normalized coordinates of points. Notably, this approach is not invariant with respect to $E(n)$, Euclidean group of in $\mathbb{R}^n$ (i.e. it is not symmetric to transformations that preserve euclidean distance such as rotations, translations, and reflections).

For experiments on synthetic data node embeddings of each convolutional layer are concatenated and followed by a global max-pooling of node embeddings with and a 2-layer MLP classifier. To perform feature extraction we remove the last linear layer of the MLP.

**E($n$)-Equivariant Neural Network (EGNN).** To overcome the limitations of GCN we also explore E(2)-equivariant networks (and thus, allowing us to get E(2)-invariant embeddings for inputs defined by sets of Cartesian coordinates in $\mathbb{R}^2$). Introduced in Satorras et al. (2021), they offer a simple recipe to enforce symmetry with respect to the E($n$) groups. The key idea rests on defining messages as functions of pairwise distances between node positions (which are preserved under E($n$) transformations) and E($n$)-invariant node features.

We define node features using local degree profiles as described in Cai & Wang (2022) which aggregate statistics of the neighbor degrees. Since the adjacency matrix of our graph is given by the Euclidean distances between the centroids of cell nuclei, these feature vectors remain invariant under rotations, translations, and reflections. The network is constructed of 3-layers of EGNN convolutional layers.

For experiments on synthetic data node embeddings of the last convolutional layer are mean-pooled and passed through a 2-layer MLP classifier. To perform feature extraction we remove the last linear layer of the MLP.

### C.4 ADAPTATION TO MULTICHANNEL IMAGES

Cell Painting images comprise five channels in JUMP-CP and six channels in RxRx3-core. As most vision backbones are pretrained on RGB images, their input projection layers must be adapted to accept $n \in \{5, 6\}$ input channels.

For ViT-based encoders, including DINO and H-Optimus, we expand the pretrained patch embedding weights to the required number of input channels using a custom channel-expansion scheme that preserves and copies the original filters and the bias when present, while for ResNet backbones we adapt the first convolutional layer using the `timm.create_model in_chans` argument, which additionally rescales the expanded weights to preserve activation magnitudes.

## D STRUCTURE-BASED TISSUE REPRESENTATIONS.

### D.1 CELL COUNT.

Benchmark scores on HEST-1k are evaluated on predictions of a single linear ridge regression layer. Since embeddings of structure-based models already contain some non-linear transformation of cell count, we manually create vectors of 16 hand-crafted features based on the initial cell count value using polynomial, trigonometric, and logarithmic functions of the cell count per patch. To avoid issues during PCA computation we tile the embeddings to the target size 256 and add a small amount ($\sigma = 0.01$) of Gaussian noise to the final embeddings. We evaluate standardized and raw cell count embeddings and select the best performing configuration. Another option provided by authors is using XGBoost (Chen & Guestrin, 2016) for regression. We provide additional results for XGBoost in table 7, which yields a slight improvement overall. The results reported in the main text we follow the PCA+ridge evaluation for consistency with other models. However, for each dataset we report the results for the best combination of hand-crafted cell-counting features, giving this baseline a slight advantage.

### D.2 CONSTRUCTION OF CELL GRAPHS AND CELL POINT SETS.

Cell point sets are defined by coordinates of nuclei centroids. We reuse segmentation results obtained with CellViT (Hörst et al., 2024) as provided with HEST-1k benchmark. The absolute coordinates of cell nuclei are linked to absolute positions of the corresponding spatial transcriptomics spots.

For the non-binned version we estimate the size of each image patch in the coordinates of the source WSI using the magnification factors and estimated pixel sizes as provided with HEST-1k. The cells are binned according to the estimated patches.

For the 1NN version, the bounding boxes of aggregated *image patches* are computed as described in B. The bounding boxes are then used to bin cell coordinates into aggregated patches. Coordinates of cell patches are rescaled to [0, 1] using the largest dimension of bounding boxes. That is for each group of cells the most distant cells from the centroid does not necessarily reach the boundary of the

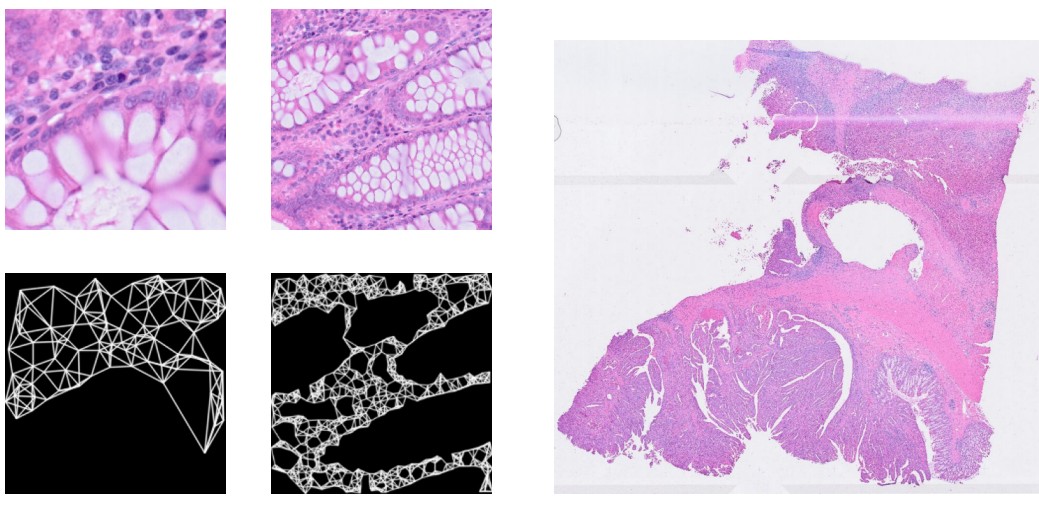

(a) 1x1 patches (left), 1NN patches (right).    (b) Slide: `TENX147`, dataset: COAD.

Figure 10: **Multiscale views in WSIs.** (a) Comparison of the original patches from HEST-1k and HEST-1k-1NN and the corresponding cell graphs. (b) The source WSI from which the samples are taken. The slide overlays a square grid of ST spots and the aggregated patch corresponds to a 9-spot neighborhood.

resulting $[0, 1] \times [0, 1]$ square. The final coordinates are centered and rescaled to approximately [-1, 1].

The binarized images of cell graphs are plots of 5-nearest-neighbor graphs. The number of neighbors is chosen based on a trade-off between reflecting variations in cellular density and following outlines of large cellular formations as shown in Fig. 10a. The graphs are rendered at the pixel count given by the dimensions of bounding boxes of patches at the resolution of the corresponding source WSIs scaled by the estimated pixel size per $\mu$m (provided with HEST-1k). This approach standardizes patch dimensions, approximately linking it to a physical ST spot across datasets imaged at different resolutions. The rendered graphs are then resized to 224×224 pixels using linear interpolation.

### D.3 SYNTHETIC DATA.

To explore effectiveness of geometric modalities for capturing relevant biological signal we design a synthetic dataset of points on the unit square. Notably we want to explore the following questions while controlling for the cell count.

First, we are interested in detections of local variations of density. Such variations are intended to represent cell clumping or large structural elements of cellular tissue e.g. ducts in gland samples. We model cell graphs using an inhomogeneous Poisson process with spatial intensity $\lambda(x, y)$ for $(x, y) \in [0, 1] \times [0, 1]$.

Second, we want to verify separability of samples defined on different grids. Drawing inspiration from works in cancer subtyping (Wang et al., 2023), which demonstrate how regularity of cell organization can help discriminate between specific pathological conditions, we additionally simulate cell graphs as adaptive square grids with the resolution of the grid given by local cell density. Cell grids are then perturbed with gaussian noise to create the final cell positions. The variance of added gaussian noise is proportional to some $\sigma(x, y)$ for $(x, y) \in [0, 1] \times [0, 1]$. While real cellular tissue is not organized in regular square grids, we are interested in the model's ability to detect localized disruptions of regular patterns.

We select density and noise patterns of interest and unify them into the following landscapes:

$$\text{SlopeLandscape}(x, y; k, b) = 1 + \max(b - kx, 0)$$
$$\text{StepLandscape}(x, y; a_x, \Delta) = 1 + \Delta \cdot \mathbf{1}_{x < a_x}.$$

We define $\text{Discs}(x, y; n, r, \text{emboss}, \Delta)$ as

$$1 + \Delta \cdot \min\left(\sum_i^n \mathbf{1}_{(x,y) \in \text{disc}_i}, 1\right).$$

Then $\text{Discs}(x, y; n, r, \text{deboss})$ is given by

$$1 - \min\left(\sum_i^n \mathbf{1}_{(x,y) \in \text{disc}_i}, 1\right).$$

The discs $\text{disc}_i$, $i = 1, \ldots, n$ have a shared radius $r$ and are uniformly spaced at a distance of $0.25$ from the center of unit square if $n > 1$ and randomly positioned within $[r, 1-r] \times [r, 1-r]$ square.

To address the first question we introduce varying numbers of discs as well as a single disc setup with the location of the disc being randomly sampled. The area of the discs is adjusted to either preserve the area of a larger reference disc or not. For the second question we use the slope and step functions as scalers for density and the amount of added noise. This leads us to the classification problem as summarized in table 4. For each class we generate a train/validation/test split of size 1000/100/1000 samples respectively. Finally we randomly sample 90-degree rotations and mirror flips for images and unconstrained random rotations for point sets both during training as data augmentations and during evaluation. From the definition of classes it is clear that some may overlap or not manifest obvious differences under uniform and noisy grid-based sampling.

A visualization of a subset of noise/ density landscapes is given in Fig. 11a. We provide binary images of graphs for each class in Fig. 11b.

We conduct three classification experiments. To evaluate the capacity of vision encoders to classify binary graphs we fully train a ResNet18 model from scratch. To evaluate capabilities of pretrained vision encoders on this task we train a linear classifier on top of a frozen ViT-l/16 pretrained on natural images. Finally, we train a GNN architecture on *k*NN graphs constructed from node positions. Table 5 summarizes our results.

While none of the models achieve a perfect score they demonstrate an above random performance and allow us to conclude that cell position sets as a modality opens a promising direction for studying tissue organization beyond cell count.

# E    COMPARISON AND ADDITIONAL EXPERIMENTAL RESULTS FOR HEST-1K AND HEST-1K-1NN.

## E.1    COMPARISON OF HEST-1K AND HEST-1K-1NN

To better understand the new level of granularity introduced with the construction of HEST-1k-1NN we provide a side-by-side comparison of source and binned patches in Fig. 10. The larger binned patches help better recognize larger structural elements in tissues (e.g. ducts in glands), while sacrificing fine resolution of cellular morphology. While this can be expected to favor strong structure-based models, the impact of the new task on pretrained foundation models is not immediately clear. Especially since both global tumoral organization and single-cell morphology can hint at increased expression of some marker genes.

We re-establish reference performance of the selected models on the two versions of the benchmark. Fig. 9 suggests a strong overall alignment between all (IID, OOD, and untrained) models. Furthermore, the new task appears to be generally easier which can be partially attributed to smoother targets. However, a closer evaluation of the foundation models specifically reveals that the final model rankings do not share the same order, as evidenced by correlation and concordance coefficients in Fig. 9b. This supports our original claim that certain dataset-specific targets are substantially easier to predict with imaging and structural modalities across all models even including naive untrained

Table 4: A Summary of Synthetic Classes.

| Class | Sampling | Noise and/ or density | Cell count |
|---|---|---|---|
| 0 | Noisy Grid, 10 voxels | $\sigma(x,y) \propto \mathrm{Id}$ | 900 |
| 1 | Noisy Grid, 10 voxels | $\sigma(x,y) \propto \mathrm{SlopeLandscape}(x,y;3,3)$ | 900 |
| 2 | Noisy Grid, 10 voxels | $\lambda(x,y) \propto \mathrm{SlopeLandscape}(x,y;3,3), \sigma(x,y) \equiv 0.01$ | 890 |
| 3 | Noisy Grid, 4 voxels | $\lambda(x,y) \propto \mathrm{StepLandscape}(x,y;0.5,1), \sigma(x,y) \equiv 0.01$ | 848 |
| 4 | Noisy Grid, 10 voxels | $\lambda(x,y) \propto \mathrm{Discs}(x,y;1,0.1,\mathrm{emboss},2), \sigma(x,y) \equiv 0.01$ | 964 |
| 5 | Noisy Grid, 10 voxels | $\lambda(x,y) \propto \mathrm{Discs}(x,y;3,0.1,\mathrm{emboss},2), \sigma(x,y) \equiv 0.01$ | 1028 |
| 6 | Noisy Grid, 10 voxels | $\lambda(x,y) \propto \mathrm{Discs}(x,y;1,0.1,\mathrm{deboss}), \sigma(x,y) \equiv 0.01$ | 864 |
| 7 | Noisy Grid, 10 voxels | $\lambda(x,y) \propto \mathrm{Discs}(x,y;3,0.1,\mathrm{deboss}), \sigma(x,y) \equiv 0.01$ | 819 |
| 8 | Uniform | $\lambda(x,y) \propto \mathrm{Discs}(x,y;3,0.1,\mathrm{emboss}),2$ | Pois(900) |
| 9 | Uniform | $\lambda(x,y) \propto \mathrm{Discs}(x,y;1,0.1,\mathrm{deboss})$ | Pois(900) |
| 10 | Uniform | $\lambda(x,y) \propto \mathrm{Id}$ | Pois(900) |
| 11 | Uniform | $\lambda(x,y) \propto \mathrm{Discs}(x,y;1,0.1,\mathrm{emboss},2)$ | Pois(900) |
| 12 | Uniform | $\lambda(x,y) \propto \mathrm{Discs}(x,y;1,0.1,\mathrm{deboss},1)$ | Pois(900) |
| 13 | Noisy Grid, 10 voxels | $\sigma(x,y) \propto \mathrm{StepLandscape}(x,y;0.5,1)$ | 900 |
| 14 | Uniform | $\lambda(x,y) \propto \mathrm{Discs}(x,y;3,\sqrt{0.2^2/3},\mathrm{emboss},2)$ | Pois(900) |
| 15 | Uniform | $\lambda(x,y) \propto \mathrm{Discs}(x,y;3,\sqrt{0.2^2/3},\mathrm{deboss})$ | Pois(900) |
| 16 | Uniform | $\lambda(x,y) \propto \mathrm{Discs}(x,y;5,\sqrt{0.2^2/5},\mathrm{emboss},2)$ | Pois(900) |
| 17 | Uniform | $\lambda(x,y) \propto \mathrm{Discs}(x,y;5,\sqrt{0.2^2/5},\mathrm{deboss})$ | Pois(900) |
| 18 | Uniform | $\lambda(x,y) \propto \mathrm{SlopeLandscape}(x,y;3,3)$ | Pois(900) |
| 19 | Uniform | $\lambda(x,y) \propto \mathrm{SlopeLandscape}(x,y;2,2)$ | Pois(900) |
| 20 | Uniform | $\lambda(x,y) \propto \mathrm{SlopeLandscape}(x,y;1,1)$ | Pois(900) |
| 21 | Uniform | $\lambda(x,y) \propto \mathrm{Discs}(x,y;1,0.2,\mathrm{emboss},2)$ | Pois(900) |
| 22 | Uniform | $\lambda(x,y) \propto \mathrm{Discs}(x,y;1,0.2,\mathrm{deboss})$ | Pois(900) |
| 23 | Uniform | $\lambda(x,y) \propto \mathrm{StepLandscape}(x,y;0.5,1)$ | Pois(900) |

Table 5: Test scores on our 24-class synthetic dataset. Classes with F1-scores above the threshold of 0.98 are provided.

| Model | Average F1 Score | Worst Class (F1-score) | Best Class (F1-score) | Invariance |
|---|---|---|---|---|
| ResNet18 (from scratch) | 0.87 | class 18 (0.38) | classes 1,3,5,7,8,9,14,15,17,19,21 ($\geq$**0.99**) | Learned, data augmentations |
| ViT-l/16 (linear probing) | 0.68 | class 22 (0.15) | classes 7,9,12,15,17,20,21 ($\geq$ **0.98**) | Learned, data augmentations |
| GCN (from scratch) | 0.72 | class 4 (0.15) | classes 7,8,9,12,15,17,21 ($\geq$**0.98**) | Learned, data augmentations |
| E2GNN (from scratch) | 0.73 | class 4 (0.04) | classes 1,5,12,17,18,19 (**=1.0**) | Architectural, E(2) |

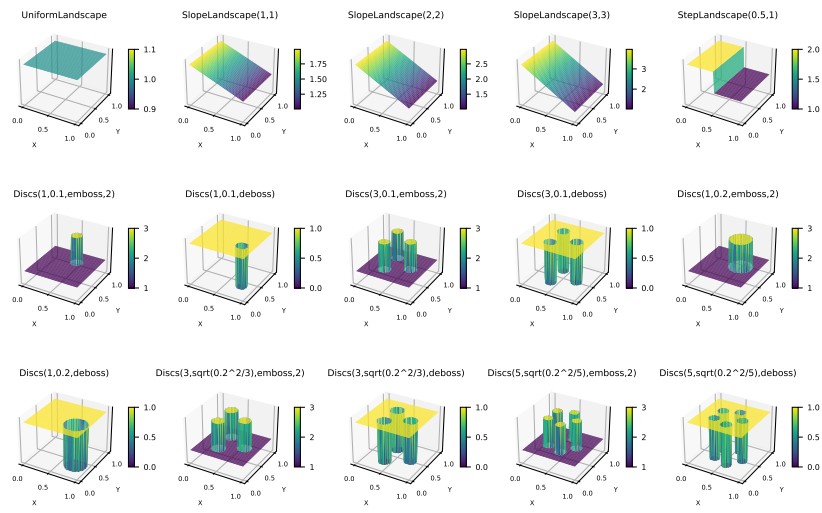

(a) Examples of intensity functions.

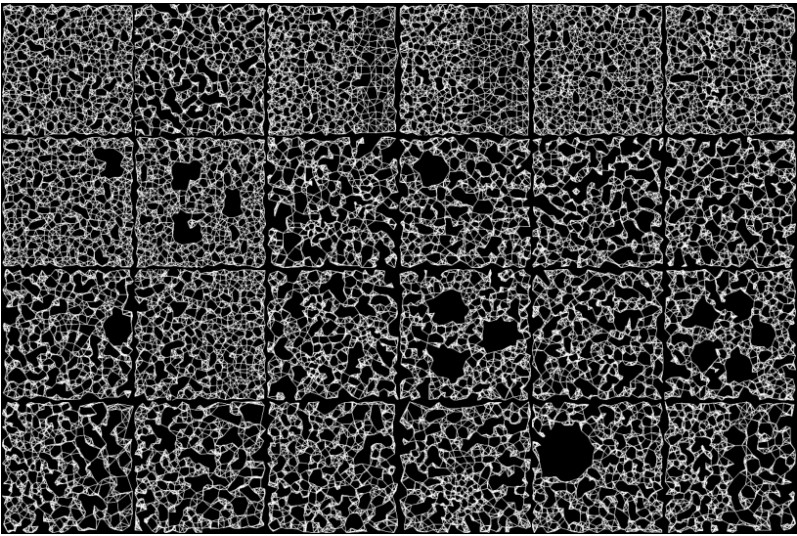

(b) Test samples from each of 24 synthetic classes, rendered at 224×224 pixels.

Figure 11: **Illustration of synthetic graph data.** Samples are designed to represent variations in local density while controlling for the node count. (a) Selected intensity functions that are used to scale density $\lambda$ and spatial noise $\sigma$ as listed in table 4. (b) Test samples for each of 24 classes.

baselines. A direct inspection of metrics in table 8 points out several datasets that preserve top-1 and top-2 rankings. For instance H-Optimus-1 successfully outperforms its competitors on SCKM under both levels of granularity. Similarly, CONCH v1.5 is a strong performer on PRAD, while H-Optimus-0 reaches the highest scores on the challenging CCRCC dataset.

## E.2 STRUCTURAL BASELINES APPLY TO THE NON-BINNED HEST-1K.

As shown in Fig. 12 the original non-binned dataset already contains a large subset of targets that can be explained by structure-only features. Notably the PCC scores for a large cluster of genes predicted with the structure-only ResNet152 aligns well with scores achieved by evaluating an IID (Fig. 12a) and OOD (Fig, 12b) baselines. This observation not only suggests that evaluation of OOD baselines alone is insufficient to properly contextualize the performance of IID models but also helps formulate hypotheses about the nature of the predictive signal captured by different pretrained encoders. For instance, when comparing DINOv2-g/14 and H-Optimus-1 on a subset of SKCM one can choose to focus specifically on the genes that are not well predicted by the structure-based model, hence restricting the analysis to as subset of targets, which appears to be more affected by cellular morphology.

## E.3 THE IMPACT OF CELL COUNT

The performance of structure-only models exceeds random levels and even occasionally reaches pretrained image-based models for both HEST-1k-1NN and HEST-1k. For the latter, larger structural patterns are less prominent, and thus it is of immediate interest to compare the models to a simple cell counting baseline. In Fig. 13 we observe that while the individual PCC values per gene may differ considerably for binary images of graphs and cell count, the overall trend indicates a strong similarity between the two approaches. In particular for the non-binned version of the benchmark, gene-wise PCC scores on COAD are strongly correlated (Fig. 13b). This result is not surprising given that for a given patch the number of cells may correlate with local density, regularity of the grid formed by a rendered graph, and other visual features captured by a pretrained CNN. Thus, a vision encoder, not trained to do so, might be indirectly estimating the number of cells in a patch. Removal of morphological features allows us to highlight this behavior in encoders pretrained on natural images.

While for a given slide cell count might be perfectly correlated with other spatial features, it is easy to construct counterexamples where discriminating between spatial arrangements requires more expressive approaches. We investigate such cases in D.3 and conclude that both GNNs and common vision encoder can recover features beyond cell count. Explicitly demonstrating their utility however remains a more complicated task, requiring a controlled setup that is difficult to achieve in context of heterogeneous tissue samples.

In table 7 we list scores of cell-counting baselines for all the subsets of the benchmark. Notably our evaluation of GNNs pretrained on synthetic data (controlled for the average cell count) yields inconsistent results: the GCN appears to benefit from such pretraining while, the opposite is true for the EGNN. In Fig. 14 we overlay dimensionality reduction plots with values of patch-wise cell count and indices of the corresponding WSIs. The UMAP plots display a gradient of cell count which appears to be one of the prominent axes of variation for both the untrained and the pretained GCNs, suggesting that training on estimation of local density patterns still yields representations that correlate strongly with cell count in real tissue samples.

## E.4 THE IMPACT OF PRETRAINING STRATEGIES ON STRUCTURE-ONLY MODELS

We investigate the contribution of different pretraining strategies on the structure-based modalities. For image encoders evaluated on binarized images of cell graphs natural image pretraining might provide some ability to capture and recognize larger shapes as well as local textural patterns, providing features potentially helpful at organ type identification and cell count. Conversely, 2D-coordinate-only models in this domain do not have reference pretraining datasets. Furthermore, using tissue patches from the non-benchmark part of HEST-1k and other open datasets requires a meaningful pre-training tasks. Because of strong correlation with simple features like cell count (Fig. 13b) pretraining on gene expression prediction might lead to inherently biased representations,

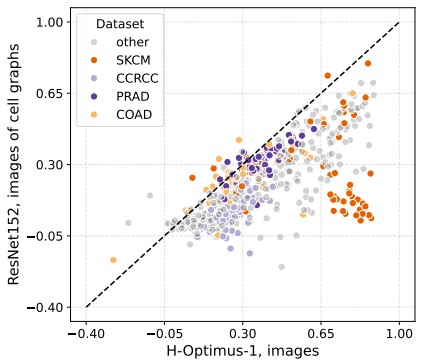
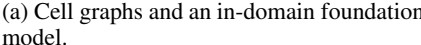

(a) Cell graphs and an in-domain foundation model.

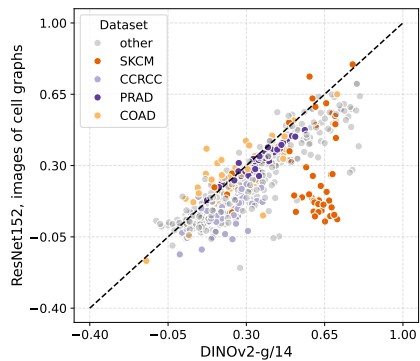

(b) Cell graphs and an out-of-domain vision backbone.

Figure 12: **Experiments on the non-binned HEST-1k.** Structure-based models remain competitive on a subset of genes across all ranges of PCC scores. (a) Gene-wise PCC for images of cell graphs and an in-domain foundation model H-optimus-1. (b) Gene-wise PCC for images of cell graphs and an out-of-domain DINOv2-g/14.

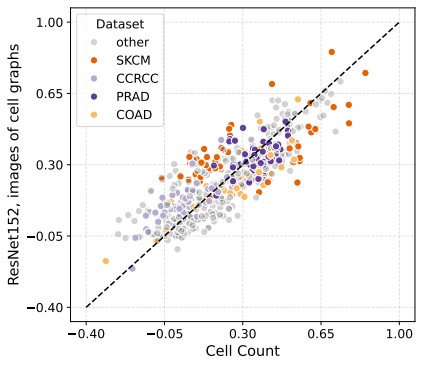
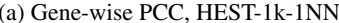

(a) Gene-wise PCC, HEST-1k-1NN.

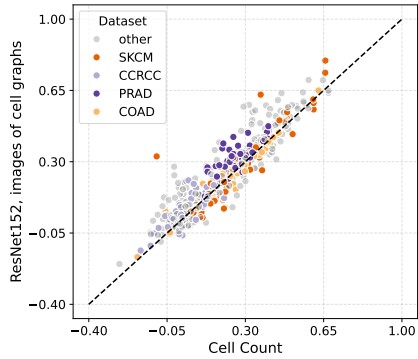

(b) Gene-wise PCC, HEST-1k.

Figure 13: **Possible contribution of cell count.** Each point represents gene-wise PCC for images of cell graphs and a non-linear cell-counting baseline. (a) Binned HEST-1k-1NN. (b) Non-binnned HEST-1k.

while development of relevant SSL methods exceeds the scope of this work. We thus choose to restrict ourselves to a classification problem on cell count-controlled synthetic dataset as described in D.3. We expect this task to allow the models to capture some basic variations in local density beyond what untrained models can offer.

Table 6 shows that both natural images and synthetic data can provide a meaningful training signal for structure encoders. It is generally generally true for image-based encoders (table 9), making this baseline easy to implement. Synthetic pretraining of GNNs yields varying results, but can be successful as with our GCN model. Yet, it is possible that the model implicitly leverages even simpler patterns like local cell count, a strong hand-crafted baseline, as discussed in Appendix E.

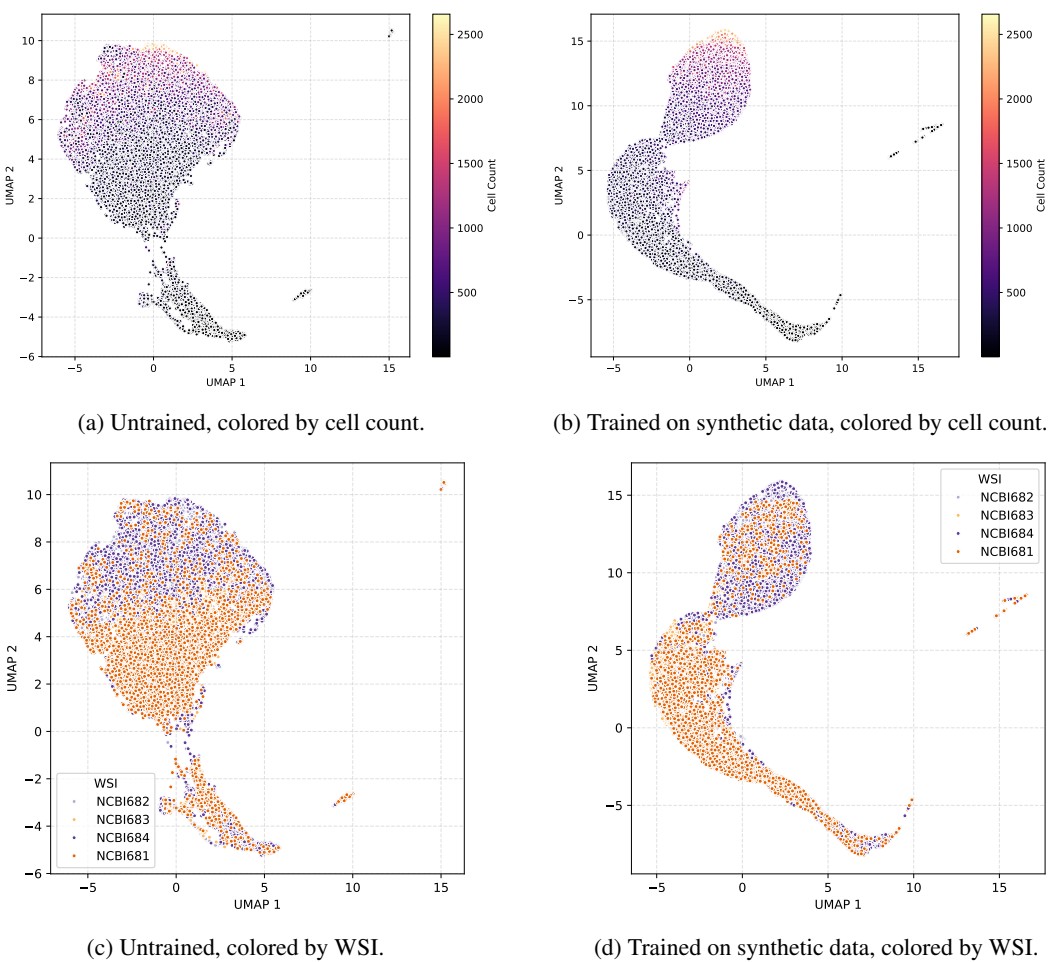

(a) Untrained, colored by cell count.

(b) Trained on synthetic data, colored by cell count.

(c) Untrained, colored by WSI.

(d) Trained on synthetic data, colored by WSI.

Figure 14: **Cell count and slide separation in GNNs.** UMAP plots of patch-level embeddings extracted with an untrained and trained GCN on LYMPH IDC-1NN.

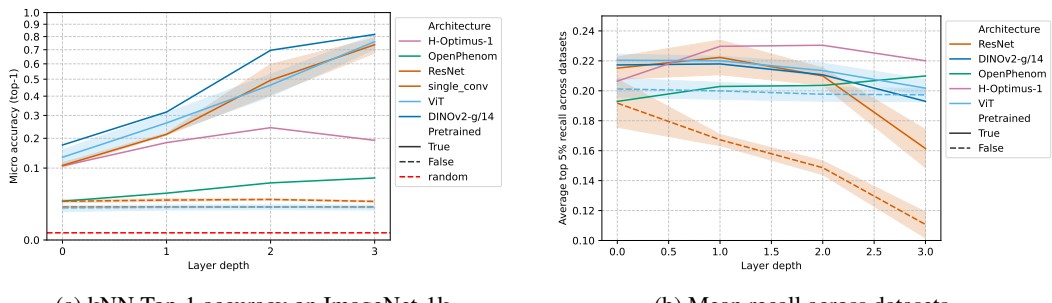

(a) kNN Top-1 accuracy on ImageNet-1k.

(b) Mean recall across datasets.

Figure 15: **Model performance as a function of network stage.** Both panels evaluate embeddings at four intermediate stages including the last layer. (a) Evolution of micro-accuracy on the ImageNet-1k validation set. (b) Average recall across all RxRx3-Core evaluated datasets.

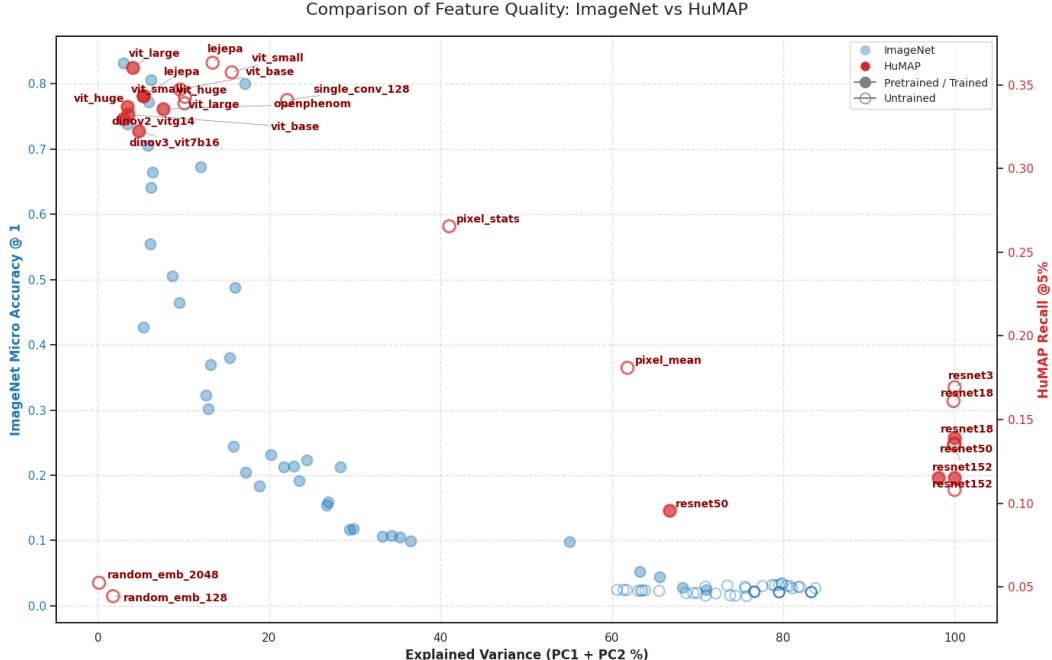

Figure 16: **Representational dimensionality versus performance on biological and natural image benchmarks**. Dataset specific scores are plotted against variance explained by the first two PCA components. Right axis: HuMAP mean recall at top 5%. Left axis: ImageNet-1k kNN top-1 accuracy. Points correspond to different model architectures, training states, and extraction layers.

# F   DETAILED MODEL PERFORMANCE ON HEST-1K AND HEST-1K-1NN

Table 6: Impact of pretraining on structure-only models on HEST-1k-1NN. We report average gene-wise PCC per dataset and the absolute improvement over the untrained version in parenthesis. Standard deviations for cell count are computed over the cross-validation folds.

| MODEL | TRAINING | CCRCC | PRAD | SKCM | COAD |
|---|---|---|---|---|---|
| CELL COUNT | NO | $0.11 \pm 0.07$ | $0.37 \pm 0.05$ | $0.40 \pm 0.06$ | $0.30 \pm 0.07$ |
| GCN | YES, SYNTH. DATA | 0.04 (-0.02) | 0.38 (+0.04) | 0.45 (+0.08) | 0.31 (+0.04) |
| EGNN | YES, SYNTH. DATA | 0.04 (-0.01) | 0.36 (+0.005) | 0.22 (-0.05) | 0.28 (-0.02) |
| RESNET152 | YES, IN1K | 0.11 (+0.04) | 0.34 (+0.09) | 0.37 (+0.23) | 0.24 (+0.14) |

Table 7: **Pearson Correlation Coefficient (PCC) across datasets for cell-counting and GNN baselines for HEST-1k and HEST-1k-1NN**. The best and the second best score for HEST-1k-1NN are in **bold** and underlined respectively. Standard deviations across the cross-validation folds are reported.

| Modality | Training | Benchmark Model | IDC | PRAD | PAAD | SKCM | COAD | READ | CCRCC | LUNG | LYMPH IDC | Avg |
|---|---|---|---|---|---|---|---|---|---|---|---|---|
| 1x1 | Hand-crafted | Cell count (PCA+ridge) | 0.32 ± 0.04 | 0.26 ± 0.02 | 0.32 ± 0.04 | 0.26 ± 0.05 | 0.26 ± 0.004 | 0.02 ± 0.001 | 0.04 ± 0.07 | 0.42 ± 0.003 | 0.11 ± 0.05 | 0.22 ± 0.13 |
| | | Cell count (XGBoost) | 0.31 ± 0.05 | 0.26 ± 0.02 | 0.34 ± 0.08 | 0.30 ± 0.07 | 0.26 ± 0.01 | 0.05 ± 0.03 | 0.03 ± 0.06 | 0.44 ± 0.01 | 0.11 ± 0.07 | 0.23 ± 0.12 |
| 1NN | Hand-crafted | Cell count (PCA+ridge) | **0.38** ± 0.06 | 0.37 ± 0.05 | 0.30 ± 0.04 | 0.32 ± 0.02 | 0.30 ± 0.07 | -0.002 ± 0.03 | **0.11** ± 0.07 | **0.49** ± 0.02 | 0.21 ± 0.11 | 0.28 ± 0.14 |
| | | Cell count (XGBoost) | 0.37 ± 0.08 | 0.36 ± 0.05 | **0.34** ± 0.07 | 0.40 ± 0.06 | 0.30 ± 0.07 | 0.03 ± 0.02 | 0.04 ± 0.09 | **0.49** ± 0.02 | 0.21 ± 0.16 | 0.28 ± 0.15 |
| | Synthetic data | EGNN | 0.22 ± 0.07 | 0.36 ± 0.09 | 0.33 ± 0.09 | 0.22 ± 0.06 | 0.28 ± 0.12 | 0.02 ± 0.06 | 0.05 ± 0.09 | 0.19 ± 0.09 | 0.17 ± 0.09 | 0.20 ± 0.12 |
| | | GCN | **0.38** ± 0.04 | **0.38** ± 0.04 | 0.33 ± 0.08 | **0.45** ± 0.07 | **0.31** ± 0.01 | **0.06** ± 0.07 | 0.04 ± 0.09 | **0.49** ± 0.03 | **0.26** ± 0.12 | **0.30** ± 0.16 |
| | Untrained | EGNN | 0.35 ± 0.02 | 0.360 ± 0.05 | 0.32 ± 0.03 | 0.27 ± 0.03 | 0.30 ± 0.003 | 0.02 ± 0.06 | 0.05 ± 0.05 | 0.42 ± 0.02 | **0.26** ± 0.10 | 0.26 ± 0.14 |
| | | GCN | 0.36 ± 0.04 | 0.34 ± 0.04 | 0.32 ± 0.03 | 0.37 ± 0.08 | 0.27 ± 0.01 | 0.02 ± 0.04 | 0.06 ± 0.05 | 0.47 ± 0.001 | 0.25 ± 0.10 | 0.27 ± 0.15 |

Table 8: **Pearson Correlation Coefficient (PCC) across datasets and foundation models for HEST-1k and HEST-1k-1NN**. The best and the second best score per modality are in **bold** and underlined respectively. Standard deviations across the cross-validation folds are reported.

| Modality | Training | Benchmark Model | IDC | PRAD | PAAD | SKCM | COAD | READ | CCRCC | LUNG | LYMPH IDC | Avg |
|---|---|---|---|---|---|---|---|---|---|---|---|---|
| 1x1-img | Foundation Model | CONCH v1 | 0.536 ± 0.084 | 0.355 ± 0.010 | 0.446 ± 0.071 | 0.579 ± 0.050 | 0.253 ± 0.008 | 0.163 ± 0.049 | 0.218 ± 0.035 | 0.531 ± 0.011 | 0.251 ± 0.042 | 0.370 ± 0.157 |
| | | CONCH v1.5 | 0.544 ± 0.085 | **0.381** ± 0.010 | 0.457 ± 0.060 | 0.554 ± 0.036 | 0.279 ± 0.013 | 0.159 ± 0.066 | 0.218 ± 0.039 | 0.550 ± 0.007 | 0.270 ± 0.054 | 0.379 ± 0.154 |
| | | CTransPath | 0.511 ± 0.053 | 0.343 ± 0.046 | 0.436 ± 0.067 | 0.512 ± 0.081 | 0.228 ± 0.057 | 0.113 ± 0.077 | 0.228 ± 0.047 | 0.503 ± 0.040 | 0.235 ± 0.048 | 0.345 ± 0.151 |
| | | GigaPath | 0.553 ± 0.073 | 0.370 ± 0.021 | 0.474 ± 0.049 | 0.556 ± 0.067 | 0.291 ± 0.025 | 0.193 ± 0.067 | 0.241 ± 0.038 | 0.544 ± 0.035 | 0.252 ± 0.052 | 0.386 ± 0.148 |
| | | H-Optimus-0 | 0.598 ± 0.084 | 0.379 ± 0.002 | 0.492 ± 0.041 | 0.655 ± 0.058 | 0.302 ± 0.002 | 0.222 ± 0.048 | **0.274** ± 0.035 | 0.561 ± 0.028 | 0.260 ± 0.043 | 0.416 ± 0.163 |
| | | H-Optimus-1 | **0.604** ± 0.078 | 0.375 ± 0.006 | 0.490 ± 0.040 | **0.665** ± 0.049 | **0.321** ± 0.017 | **0.239** ± 0.038 | 0.256 ± 0.012 | **0.577** ± 0.013 | **0.277** ± 0.043 | **0.423** ± 0.164 |
| | | Hibou Large | 0.569 ± 0.079 | 0.304 ± 0.024 | 0.467 ± 0.081 | 0.588 ± 0.042 | 0.298 ± 0.035 | 0.196 ± 0.055 | 0.271 ± 0.068 | 0.575 ± 0.004 | 0.238 ± 0.042 | 0.390 ± 0.159 |
| | | Kaiko Base 8 | 0.560 ± 0.075 | 0.361 ± 0.022 | 0.458 ± 0.069 | 0.574 ± 0.065 | 0.274 ± 0.023 | 0.156 ± 0.084 | 0.231 ± 0.050 | 0.514 ± 0.037 | 0.227 ± 0.031 | 0.373 ± 0.158 |
| | | Phikon | 0.533 ± 0.091 | 0.342 ± 0.077 | 0.443 ± 0.070 | 0.539 ± 0.055 | 0.257 ± 0.007 | 0.153 ± 0.083 | 0.242 ± 0.026 | 0.550 ± 0.002 | 0.237 ± 0.046 | 0.366 ± 0.153 |
| | | Phikon v2 | 0.538 ± 0.073 | 0.353 ± 0.003 | 0.444 ± 0.063 | 0.553 ± 0.036 | 0.247 ± 0.020 | 0.177 ± 0.058 | 0.267 ± 0.036 | 0.538 ± 0.013 | 0.246 ± 0.048 | 0.374 ± 0.147 |
| | | UNI | 0.572 ± 0.083 | 0.311 ± 0.078 | 0.479 ± 0.076 | 0.624 ± 0.035 | 0.262 ± 0.031 | 0.180 ± 0.046 | 0.245 ± 0.038 | 0.552 ± 0.016 | 0.258 ± 0.044 | 0.387 ± 0.169 |
| | | UNIv2 | 0.591 ± 0.082 | 0.359 ± 0.042 | 0.502 ± 0.046 | 0.663 ± 0.015 | 0.314 ± 0.010 | 0.216 ± 0.043 | 0.266 ± 0.038 | 0.562 ± 0.010 | 0.274 ± 0.040 | 0.416 ± 0.165 |
| | | Virchow | 0.585 ± 0.093 | 0.334 ± 0.001 | **0.511** ± 0.059 | 0.621 ± 0.062 | 0.305 ± 0.005 | 0.201 ± 0.046 | 0.260 ± 0.033 | 0.567 ± 0.024 | 0.258 ± 0.042 | 0.405 ± 0.164 |
| | | Virchow 2 | 0.594 ± 0.086 | 0.356 ± 0.035 | 0.476 ± 0.068 | 0.644 ± 0.036 | 0.258 ± 0.033 | 0.203 ± 0.054 | 0.267 ± 0.049 | 0.572 ± 0.017 | 0.261 ± 0.038 | 0.403 ± 0.170 |
| 1NN-img | Foundation Model | CONCH v1 | 0.590 ± 0.071 | 0.414 ± 0.003 | 0.463 ± 0.080 | 0.612 ± 0.075 | 0.320 ± 0.020 | 0.201 ± 0.127 | 0.310 ± 0.055 | 0.550 ± 0.026 | 0.361 ± 0.066 | 0.425 ± 0.141 |
| | | CONCH v1.5 | 0.598 ± 0.080 | **0.490** ± 0.019 | 0.488 ± 0.049 | 0.596 ± 0.060 | 0.337 ± 0.038 | 0.204 ± 0.139 | 0.280 ± 0.056 | 0.589 ± 0.008 | **0.383** ± 0.062 | 0.441 ± 0.147 |
| | | CTransPath | 0.577 ± 0.070 | 0.447 ± 0.021 | 0.475 ± 0.064 | 0.574 ± 0.075 | 0.275 ± 0.104 | 0.150 ± 0.167 | 0.268 ± 0.090 | 0.531 ± 0.013 | 0.317 ± 0.093 | 0.402 ± 0.154 |
| | | GigaPath | 0.611 ± 0.069 | 0.455 ± 0.033 | 0.497 ± 0.069 | 0.652 ± 0.033 | 0.338 ± 0.047 | 0.181 ± 0.120 | 0.314 ± 0.074 | 0.561 ± 0.039 | 0.339 ± 0.080 | 0.439 ± 0.156 |
| | | H-Optimus-0 | 0.624 ± 0.055 | 0.378 ± 0.102 | 0.491 ± 0.018 | 0.641 ± 0.038 | 0.365 ± 0.042 | 0.173 ± 0.164 | **0.366** ± 0.058 | 0.580 ± 0.008 | 0.339 ± 0.053 | 0.440 ± 0.155 |
| | | H-Optimus-1 | 0.622 ± 0.070 | 0.436 ± 0.052 | 0.497 ± 0.063 | **0.675** ± 0.026 | 0.350 ± 0.037 | 0.210 ± 0.129 | 0.273 ± 0.033 | 0.596 ± 0.024 | 0.377 ± 0.028 | 0.448 ± 0.161 |
| | | Hibou Large | 0.623 ± 0.068 | 0.420 ± 0.025 | 0.484 ± 0.077 | 0.642 ± 0.012 | 0.342 ± 0.114 | 0.229 ± 0.166 | 0.314 ± 0.051 | 0.589 ± 0.022 | 0.329 ± 0.083 | 0.441 ± 0.150 |
| | | Kaiko Base 8 | 0.609 ± 0.053 | 0.440 ± 0.044 | 0.437 ± 0.059 | 0.600 ± 0.004 | 0.329 ± 0.069 | 0.163 ± 0.194 | 0.317 ± 0.067 | 0.533 ± 0.004 | 0.323 ± 0.044 | 0.417 ± 0.148 |
| | | Phikon | 0.555 ± 0.073 | 0.408 ± 0.042 | 0.406 ± 0.056 | 0.605 ± 0.042 | 0.280 ± 0.025 | 0.086 ± 0.148 | 0.299 ± 0.072 | 0.504 ± 0.054 | 0.316 ± 0.085 | 0.384 ± 0.160 |
| | | Phikon v2 | 0.566 ± 0.045 | 0.339 ± 0.115 | 0.420 ± 0.035 | 0.591 ± 0.041 | 0.262 ± 0.059 | 0.131 ± 0.143 | 0.304 ± 0.052 | 0.531 ± 0.042 | 0.316 ± 0.069 | 0.384 ± 0.154 |
| | | UNI | 0.617 ± 0.078 | 0.394 ± 0.090 | 0.478 ± 0.033 | 0.633 ± 0.041 | 0.249 ± 0.019 | 0.205 ± 0.113 | 0.290 ± 0.051 | 0.586 ± 0.017 | 0.350 ± 0.062 | 0.422 ± 0.163 |
| | | UNIv2 | 0.645 ± 0.063 | 0.449 ± 0.044 | 0.488 ± 0.047 | 0.661 ± 0.018 | **0.377** ± 0.062 | **0.310** ± 0.053 | 0.343 ± 0.027 | 0.586 ± 0.027 | 0.356 ± 0.044 | **0.468** ± 0.135 |
| | | Virchow | 0.597 ± 0.081 | 0.451 ± 0.002 | 0.481 ± 0.062 | 0.549 ± 0.049 | 0.331 ± 0.055 | 0.174 ± 0.156 | 0.268 ± 0.081 | 0.592 ± 0.032 | 0.327 ± 0.083 | 0.419 ± 0.151 |
| | | Virchow 2 | **0.651** ± 0.077 | 0.408 ± 0.086 | **0.507** ± 0.037 | 0.655 ± 0.016 | 0.305 ± 0.114 | 0.254 ± 0.132 | 0.323 ± 0.082 | **0.596** ± 0.031 | 0.374 ± 0.051 | 0.453 ± 0.154 |

Table 9: **Pearson Correlation Coefficient (PCC) across datasets and `timm` encoders for HEST-1k and HEST-1k-1NN**. The best and the second best score per dataset are in **bold** and underlined respectively. Standard deviations across the cross-validation folds are reported.

| Modality | Training | Benchmark Model | IDC | PRAD | PAAD | SKCM | COAD | READ | CCRCC | LUNG | LYMPH IDC | Avg |
|---|---|---|---|---|---|---|---|---|---|---|---|---|
| 1NN-graph | Pretrained | ResNet152 | 0.340 ± 0.036 | 0.338 ± 0.017 | 0.312 ± 0.072 | 0.366 ± 0.091 | 0.243 ± 0.040 | 0.090 ± 0.058 | 0.111 ± 0.030 | 0.382 ± 0.000 | 0.162 ± 0.042 | 0.261 ± 0.113 |
| | | ResNet18 | 0.338 ± 0.041 | 0.324 ± 0.014 | 0.295 ± 0.060 | 0.333 ± 0.066 | 0.231 ± 0.040 | 0.093 ± 0.062 | 0.109 ± 0.035 | 0.380 ± 0.002 | 0.153 ± 0.055 | 0.251 ± 0.108 |
| | | ResNet34 | 0.347 ± 0.044 | 0.327 ± 0.017 | 0.307 ± 0.072 | 0.315 ± 0.059 | 0.219 ± 0.031 | 0.068 ± 0.046 | 0.115 ± 0.033 | 0.371 ± 0.023 | 0.154 ± 0.047 | 0.247 ± 0.111 |
| | | ResNet50 | 0.341 ± 0.032 | 0.331 ± 0.015 | 0.319 ± 0.067 | 0.361 ± 0.117 | 0.212 ± 0.013 | 0.097 ± 0.051 | 0.116 ± 0.032 | 0.359 ± 0.030 | 0.157 ± 0.044 | 0.255 ± 0.109 |
| | | ViT-b/16 | 0.326 ± 0.031 | 0.339 ± 0.017 | 0.278 ± 0.043 | 0.302 ± 0.063 | 0.198 ± 0.002 | 0.110 ± 0.031 | 0.107 ± 0.026 | 0.363 ± 0.007 | 0.149 ± 0.048 | 0.241 ± 0.101 |
| | | ViT-h/14 | 0.331 ± 0.029 | 0.336 ± 0.015 | 0.296 ± 0.043 | 0.338 ± 0.068 | 0.236 ± 0.036 | 0.080 ± 0.059 | 0.100 ± 0.027 | 0.384 ± 0.007 | 0.147 ± 0.050 | 0.250 ± 0.114 |
| | | ViT-l/16 | 0.335 ± 0.033 | 0.337 ± 0.013 | 0.307 ± 0.044 | 0.306 ± 0.052 | 0.192 ± 0.003 | 0.118 ± 0.042 | 0.113 ± 0.028 | 0.373 ± 0.008 | 0.157 ± 0.046 | 0.249 ± 0.103 |
| | | ViT-s/16 | 0.331 ± 0.032 | 0.336 ± 0.016 | 0.283 ± 0.047 | 0.280 ± 0.078 | 0.200 ± 0.006 | 0.099 ± 0.050 | 0.104 ± 0.030 | 0.355 ± 0.019 | 0.148 ± 0.047 | 0.237 ± 0.102 |
| | Untrained | ResNet152 | 0.316 ± 0.038 | 0.254 ± 0.024 | 0.245 ± 0.025 | 0.139 ± 0.045 | 0.101 ± 0.044 | 0.040 ± 0.020 | 0.067 ± 0.030 | 0.247 ± 0.136 | 0.103 ± 0.079 | 0.168 ± 0.098 |
| | | ResNet18 | 0.320 ± 0.038 | 0.257 ± 0.025 | 0.275 ± 0.067 | 0.157 ± 0.056 | 0.176 ± 0.027 | 0.050 ± 0.031 | 0.082 ± 0.037 | 0.384 ± 0.001 | 0.148 ± 0.069 | 0.205 ± 0.111 |
| | | ResNet34 | 0.319 ± 0.037 | 0.255 ± 0.024 | 0.284 ± 0.061 | 0.169 ± 0.050 | 0.169 ± 0.021 | 0.055 ± 0.034 | 0.082 ± 0.035 | 0.368 ± 0.007 | 0.147 ± 0.068 | 0.205 ± 0.107 |
| | | ResNet50 | 0.316 ± 0.038 | 0.253 ± 0.025 | 0.282 ± 0.060 | 0.057 ± 0.024 | 0.175 ± 0.027 | 0.045 ± 0.028 | 0.066 ± 0.031 | 0.285 ± 0.095 | 0.144 ± 0.069 | 0.180 ± 0.108 |
| | | ViT-b/16 | 0.320 ± 0.052 | 0.277 ± 0.019 | 0.280 ± 0.054 | 0.303 ± 0.074 | 0.185 ± 0.027 | 0.082 ± 0.058 | 0.079 ± 0.019 | 0.390 ± 0.004 | 0.153 ± 0.060 | 0.230 ± 0.110 |
| | | ViT-h/14 | 0.320 ± 0.054 | 0.161 ± 0.140 | 0.271 ± 0.058 | 0.341 ± 0.094 | 0.190 ± 0.026 | 0.079 ± 0.063 | 0.071 ± 0.022 | 0.189 ± 0.145 | 0.147 ± 0.055 | 0.197 ± 0.097 |
| | | ViT-l/16 | 0.318 ± 0.055 | 0.277 ± 0.020 | 0.276 ± 0.058 | 0.337 ± 0.091 | 0.185 ± 0.028 | 0.082 ± 0.061 | 0.079 ± 0.022 | 0.383 ± 0.003 | 0.150 ± 0.060 | 0.232 ± 0.112 |
| | | ViT-s/16 | 0.321 ± 0.051 | 0.278 ± 0.020 | 0.275 ± 0.053 | 0.315 ± 0.090 | 0.188 ± 0.032 | 0.080 ± 0.060 | 0.081 ± 0.023 | 0.388 ± 0.007 | 0.149 ± 0.063 | 0.230 ± 0.111 |
| 1NN-img | Pretrained | ResNet152 | 0.487 ± 0.033 | 0.385 ± 0.040 | 0.393 ± 0.043 | 0.521 ± 0.054 | 0.252 ± 0.096 | 0.110 ± 0.099 | 0.216 ± 0.057 | 0.488 ± 0.012 | **0.324** ± 0.062 | 0.353 ± 0.140 |
| | | ResNet18 | 0.475 ± 0.034 | 0.409 ± 0.009 | 0.428 ± 0.049 | 0.409 ± 0.090 | 0.264 ± 0.096 | 0.108 ± 0.111 | 0.263 ± 0.071 | 0.501 ± 0.016 | 0.264 ± 0.070 | 0.347 ± 0.129 |
| | | ResNet34 | 0.490 ± 0.034 | 0.320 ± 0.082 | 0.392 ± 0.049 | 0.417 ± 0.018 | 0.268 ± 0.037 | 0.095 ± 0.119 | 0.257 ± 0.060 | 0.448 ± 0.081 | 0.303 ± 0.064 | 0.332 ± 0.120 |
| | | ResNet50 | 0.504 ± 0.015 | **0.410** ± 0.014 | 0.387 ± 0.055 | 0.505 ± 0.076 | 0.297 ± 0.104 | 0.122 ± 0.095 | **0.264** ± 0.069 | 0.482 ± 0.026 | 0.294 ± 0.074 | 0.363 ± 0.130 |
| | | ViT-b/16 | 0.509 ± 0.045 | 0.391 ± 0.027 | 0.392 ± 0.064 | 0.510 ± 0.000 | 0.195 ± 0.096 | 0.122 ± 0.062 | 0.216 ± 0.059 | 0.475 ± 0.001 | 0.317 ± 0.055 | 0.347 ± 0.144 |
| | | ViT-h/14 | 0.509 ± 0.035 | 0.391 ± 0.001 | 0.426 ± 0.047 | 0.530 ± 0.071 | 0.275 ± 0.117 | **0.174** ± 0.098 | 0.259 ± 0.062 | 0.503 ± 0.028 | 0.301 ± 0.064 | 0.374 ± 0.128 |
| | | ViT-l/16 | **0.544** ± 0.046 | 0.365 ± 0.040 | **0.428** ± 0.069 | **0.617** ± 0.074 | 0.298 ± 0.069 | 0.107 ± 0.119 | 0.245 ± 0.050 | **0.535** ± 0.008 | 0.315 ± 0.081 | **0.384** ± 0.163 |
| | | ViT-s/16 | 0.499 ± 0.049 | 0.356 ± 0.032 | 0.378 ± 0.047 | 0.469 ± 0.016 | 0.287 ± 0.084 | 0.100 ± 0.073 | 0.190 ± 0.032 | 0.468 ± 0.028 | 0.310 ± 0.064 | 0.340 ± 0.134 |
| | Untrained | ResNet152 | 0.327 ± 0.034 | 0.232 ± 0.053 | 0.298 ± 0.090 | 0.299 ± 0.044 | 0.019 ± 0.009 | 0.039 ± 0.051 | 0.115 ± 0.139 | 0.366 ± 0.068 | 0.182 ± 0.085 | 0.209 ± 0.127 |
| | | ResNet18 | 0.317 ± 0.071 | 0.227 ± 0.055 | 0.284 ± 0.100 | 0.208 ± 0.135 | 0.152 ± 0.013 | 0.074 ± 0.096 | 0.089 ± 0.134 | 0.363 ± 0.036 | 0.199 ± 0.118 | 0.212 ± 0.098 |
| | | ResNet34 | 0.284 ± 0.081 | 0.266 ± 0.086 | 0.311 ± 0.079 | 0.297 ± 0.095 | 0.091 ± 0.094 | 0.045 ± 0.059 | 0.122 ± 0.137 | 0.413 ± 0.107 | 0.173 ± 0.103 | 0.222 ± 0.121 |
| | | ResNet50 | 0.275 ± 0.097 | 0.190 ± 0.072 | 0.276 ± 0.057 | 0.109 ± 0.094 | 0.063 ± 0.076 | 0.056 ± 0.068 | 0.068 ± 0.127 | 0.273 ± 0.058 | 0.142 ± 0.098 | 0.161 ± 0.095 |
| | | ViT-b/16 | 0.282 ± 0.125 | 0.253 ± 0.056 | 0.302 ± 0.048 | 0.316 ± 0.067 | 0.233 ± 0.019 | 0.095 ± 0.035 | 0.104 ± 0.127 | 0.414 ± 0.037 | 0.183 ± 0.119 | 0.242 ± 0.103 |
| | | ViT-h/14 | 0.239 ± 0.153 | 0.194 ± 0.029 | 0.312 ± 0.048 | 0.338 ± 0.055 | 0.229 ± 0.035 | 0.105 ± 0.043 | 0.101 ± 0.124 | 0.407 ± 0.058 | 0.187 ± 0.111 | 0.234 ± 0.103 |
| | | ViT-l/16 | 0.280 ± 0.124 | 0.216 ± 0.030 | 0.307 ± 0.048 | 0.311 ± 0.071 | 0.236 ± 0.038 | 0.098 ± 0.031 | 0.100 ± 0.131 | 0.421 ± 0.041 | 0.183 ± 0.118 | 0.239 ± 0.104 |
| | | ViT-s/16 | 0.286 ± 0.112 | 0.231 ± 0.049 | 0.297 ± 0.047 | 0.319 ± 0.057 | 0.267 ± 0.048 | 0.102 ± 0.035 | 0.093 ± 0.133 | 0.382 ± 0.044 | 0.183 ± 0.111 | 0.240 ± 0.098 |

# G   IMPACT OF WEIGHTS INITIALISATION ON UNTRAINED MODELS

Table 10: **Impact of weight initialization on recall at 5% across biological benchmarks.** Mean and standard deviation across 3 random seeds for untrained models. Results are reported for all benchmarks (CORUM, HuMAP, Reactome, SIGNOR, StringDB) across three evaluation folds.

| Model | Architecture | Fold | CORUM | HuMAP | Reactome | SIGNOR | StringDB |
|---|---|---|---|---|---|---|---|
| SingleConv | CNN | 1 | 0.284±0.009 | 0.363±0.007 | 0.080±0.001 | 0.037±0.008 | 0.222±0.003 |
| SingleConv | CNN | 2 | 0.281±0.021 | 0.350±0.030 | 0.064±0.001 | 0.066±0.005 | 0.226±0.011 |
| SingleConv | CNN | 3 | 0.263±0.004 | 0.309±0.014 | 0.068±0.001 | 0.046±0.005 | 0.204±0.005 |
| AlexNet | CNN | 1 | 0.111±0.039 | 0.106±0.029 | 0.059±0.005 | 0.057±0.003 | 0.087±0.016 |
| AlexNet | CNN | 2 | 0.114±0.012 | 0.120±0.006 | 0.059±0.017 | 0.057±0.013 | 0.102±0.015 |
| AlexNet | CNN | 3 | 0.103±0.023 | 0.098±0.023 | 0.049±0.010 | 0.058±0.018 | 0.089±0.021 |
| ResNet18 | CNN | 1 | 0.143±0.024 | 0.166±0.044 | 0.054±0.007 | 0.053±0.015 | 0.124±0.015 |
| ResNet18 | CNN | 2 | 0.138±0.011 | 0.156±0.008 | 0.057±0.005 | 0.055±0.004 | 0.117±0.007 |
| ResNet18 | CNN | 3 | 0.148±0.011 | 0.162±0.013 | 0.060±0.011 | 0.059±0.015 | 0.119±0.004 |
| ResNet50 | CNN | 1 | 0.122±0.018 | 0.131±0.023 | 0.055±0.008 | 0.046±0.013 | 0.100±0.007 |
| ResNet50 | CNN | 2 | 0.139±0.014 | 0.142±0.011 | 0.054±0.012 | 0.047±0.005 | 0.108±0.005 |
| ResNet50 | CNN | 3 | 0.128±0.022 | 0.134±0.029 | 0.069±0.012 | 0.059±0.014 | 0.102±0.014 |
| ResNet152 | CNN | 1 | 0.124±0.007 | 0.135±0.004 | 0.059±0.008 | 0.044±0.001 | 0.107±0.009 |
| ResNet152 | CNN | 2 | 0.133±0.034 | 0.141±0.026 | 0.054±0.010 | 0.048±0.007 | 0.109±0.016 |
| ResNet152 | CNN | 3 | 0.119±0.012 | 0.129±0.016 | 0.057±0.004 | 0.055±0.014 | 0.098±0.007 |
| ViT-s/16 | ViT | 1 | 0.344±0.011 | 0.396±0.014 | 0.056±0.001 | 0.043±0.002 | 0.251±0.011 |
| ViT-s/16 | ViT | 2 | 0.325±0.010 | 0.376±0.012 | 0.050±0.006 | 0.043±0.002 | 0.242±0.005 |
| ViT-s/16 | ViT | 3 | 0.294±0.007 | 0.300±0.002 | 0.051±0.003 | 0.044±0.003 | 0.216±0.005 |
| ViT-b/16 | ViT | 1 | 0.329±0.001 | 0.382±0.009 | 0.058±0.006 | 0.038±0.005 | 0.238±0.003 |
| ViT-b/16 | ViT | 2 | 0.318±0.002 | 0.371±0.006 | 0.052±0.004 | 0.042±0.009 | 0.242±0.004 |
| ViT-b/16 | ViT | 3 | 0.283±0.011 | 0.288±0.006 | 0.050±0.001 | 0.044±0.002 | 0.211±0.005 |
| ViT-l/16 | ViT | 1 | 0.319±0.006 | 0.372±0.013 | 0.065±0.003 | 0.033±0.002 | 0.237±0.007 |
| ViT-l/16 | ViT | 2 | 0.314±0.009 | 0.363±0.006 | 0.053±0.005 | 0.043±0.005 | 0.235±0.007 |
| ViT-l/16 | ViT | 3 | 0.276±0.006 | 0.282±0.009 | 0.061±0.004 | 0.051±0.004 | 0.207±0.005 |
| ViT-h/14 | ViT | 1 | 0.314±0.003 | 0.373±0.010 | 0.062±0.004 | 0.032±0.002 | 0.239±0.003 |
| ViT-h/14 | ViT | 2 | 0.314±0.005 | 0.368±0.009 | 0.065±0.003 | 0.055±0.005 | 0.243±0.008 |
| ViT-h/14 | ViT | 3 | 0.270±0.004 | 0.287±0.003 | 0.059±0.001 | 0.052±0.002 | 0.212±0.008 |

Table 11: **Per-fold compound retrieval performance on the JUMP-CP subset benchmark**. Mean mAP is reported as mean $\pm$ standard deviation across 3 random seeds.

| Model | Architecture | Fold | Mean mAP |
|---|---|---|---|
| SingleConv | CNN | 0 | 0.357±0.018 |
| SingleConv | CNN | 1 | 0.315±0.008 |
| SingleConv | CNN | 2 | 0.366±0.017 |
| SingleConv | CNN | 3 | 0.340±0.016 |
| SingleConv | CNN | 4 | 0.341±0.023 |
| ResNet18 | CNN | 0 | 0.146±0.002 |
| ResNet18 | CNN | 1 | 0.252±0.006 |
| ResNet18 | CNN | 2 | 0.293±0.011 |
| ResNet18 | CNN | 3 | 0.272±0.007 |
| ResNet18 | CNN | 4 | 0.251±0.007 |
| ResNet50 | CNN | 0 | 0.152±0.000 |
| ResNet50 | CNN | 1 | 0.275±0.007 |
| ResNet50 | CNN | 2 | 0.332±0.007 |
| ResNet50 | CNN | 3 | 0.303±0.011 |
| ResNet50 | CNN | 4 | 0.274±0.006 |
| ResNet152 | CNN | 0 | 0.313±0.006 |
| ResNet152 | CNN | 1 | 0.274±0.001 |
| ResNet152 | CNN | 2 | 0.329±0.002 |
| ResNet152 | CNN | 3 | 0.302±0.003 |
| ResNet152 | CNN | 4 | 0.300±0.003 |
| ViT-s/16 | ViT | 0 | 0.313±0.007 |
| ViT-s/16 | ViT | 1 | 0.277±0.005 |
| ViT-s/16 | ViT | 2 | 0.336±0.005 |
| ViT-s/16 | ViT | 3 | 0.306±0.002 |
| ViT-s/16 | ViT | 4 | 0.298±0.006 |
| ViT-b/16 | ViT | 0 | 0.328±0.006 |
| ViT-b/16 | ViT | 1 | 0.287±0.004 |
| ViT-b/16 | ViT | 2 | 0.352±0.005 |
| ViT-b/16 | ViT | 3 | 0.323±0.006 |
| ViT-b/16 | ViT | 4 | 0.310±0.001 |
| ViT-l/16 | ViT | 0 | 0.331±0.002 |
| ViT-l/16 | ViT | 1 | 0.289±0.000 |
| ViT-l/16 | ViT | 2 | 0.357±0.002 |
| ViT-l/16 | ViT | 3 | 0.326±0.003 |
| ViT-l/16 | ViT | 4 | 0.316±0.004 |
| ViT-h/14 | ViT | 0 | 0.354±0.003 |
| ViT-h/14 | ViT | 1 | 0.310±0.005 |
| ViT-h/14 | ViT | 2 | 0.381±0.006 |
| ViT-h/14 | ViT | 3 | 0.343±0.003 |
| ViT-h/14 | ViT | 4 | 0.337±0.006 |

# H  ADDITIONAL RESULTS ON RXRX3-CORE.

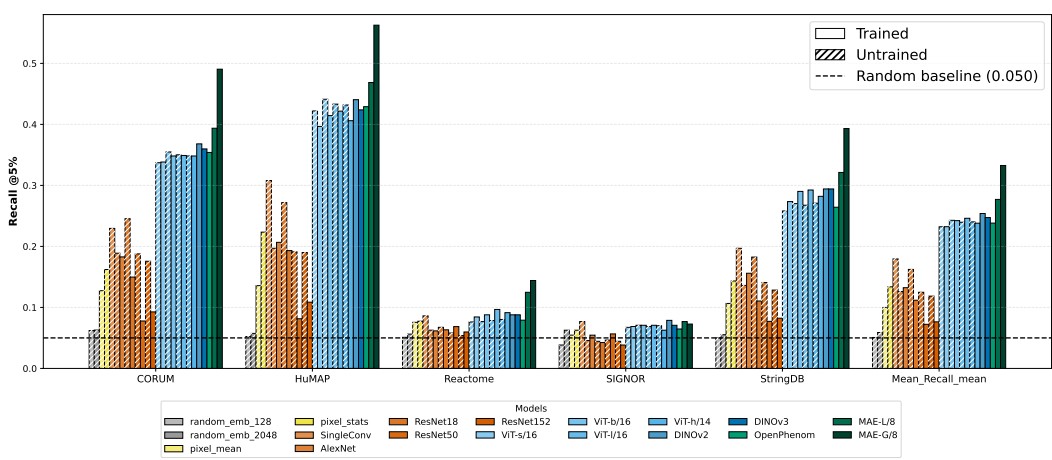

Figure 17: **Recall @5% for all models on the original full RxRx3-Core dataset.**

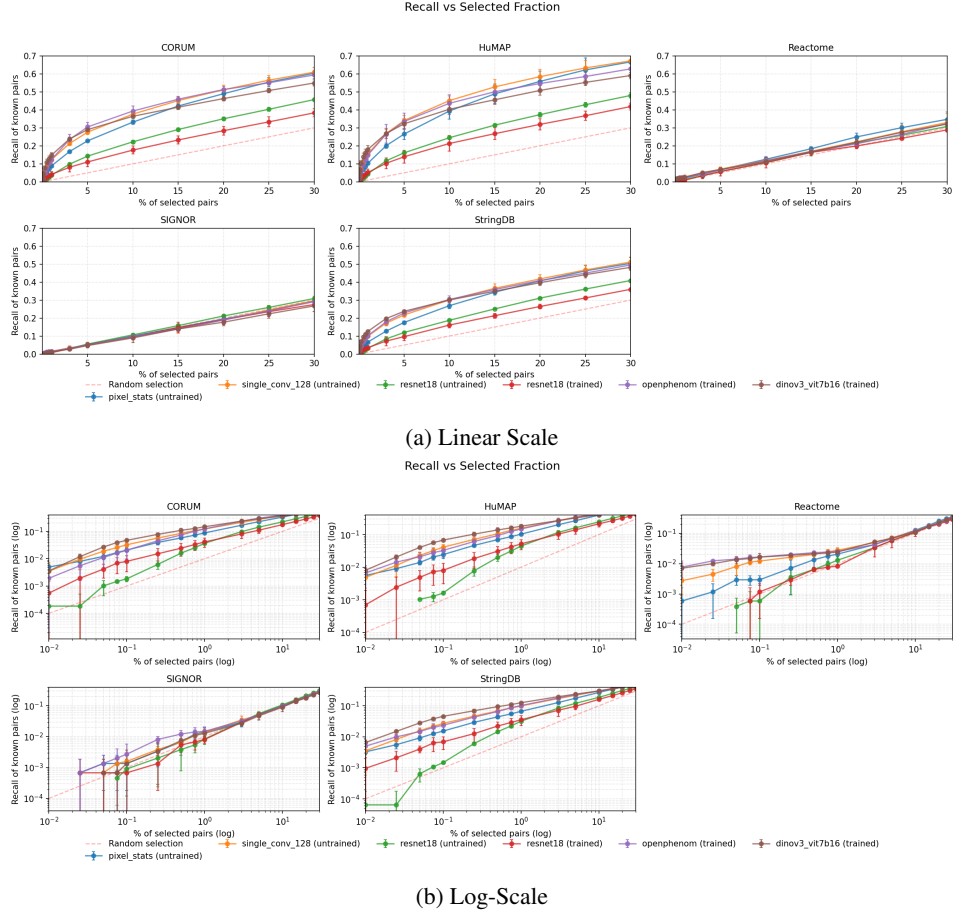

(a) Linear Scale

(b) Log-Scale

Figure 18: **Impact of selected pairs % on the recall per dataset.** A few important and diverse models are displayed. Red dash line indicates the random selection

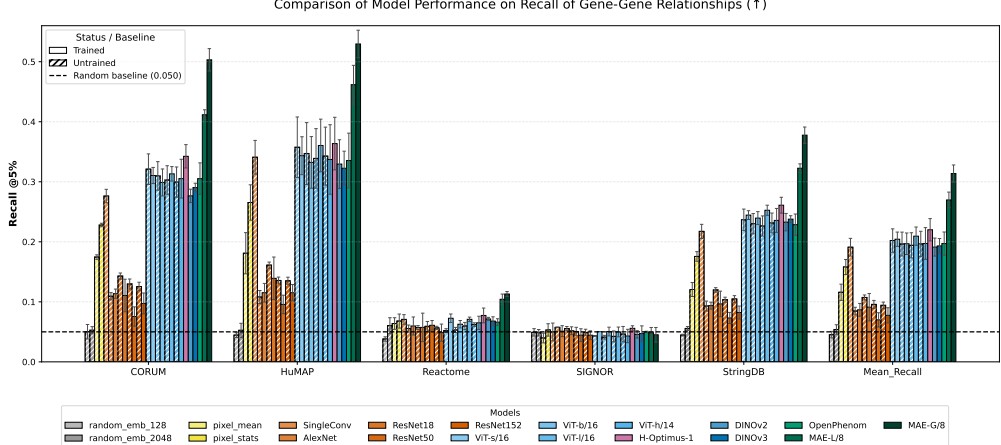

(a) Recall at 5% performance across multiple gene interaction benchmarks (CORUM, HuMAP, Reactome, SIGNOR, StringDB).

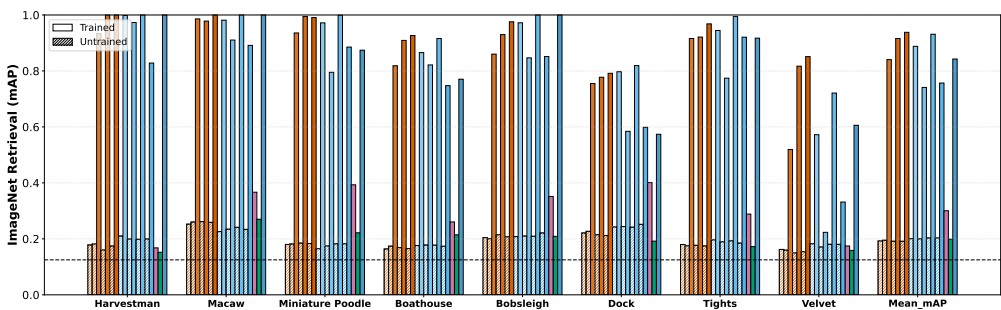

(b) Evaluation of mAP on ImageNet-1k validation set.

Figure 19: **Benchmarking model performance across cell culture and natural images tasks.** Performance is evaluated across: (a) gene-gene interaction retrieval and (b) ImageNet-1k classification. Solid bars indicate pretrained models; hatched bars indicate untrained models. Error bars represent variability across evaluation folds, and dashed horizontal lines denote random baselines.

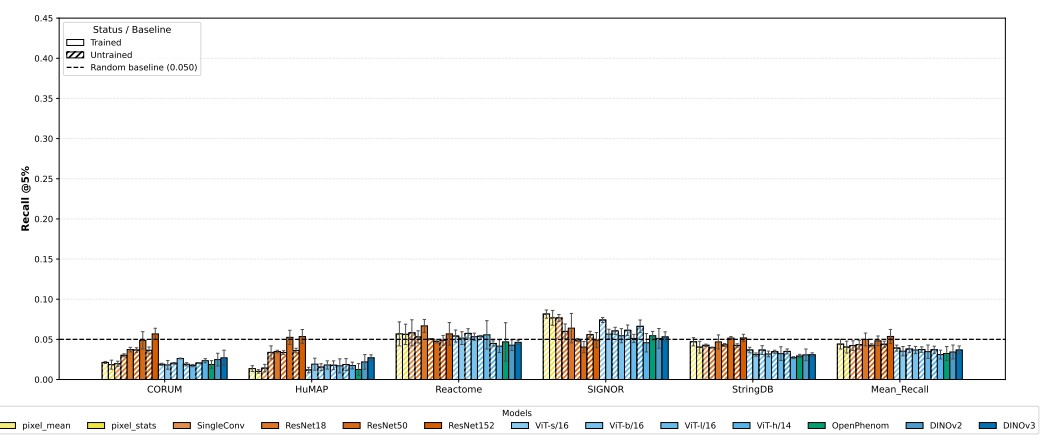

Figure 20: Model performance on gene-gene interaction retrieval when looking at bottom 5% of cosine similarities between gene-gene pairs.

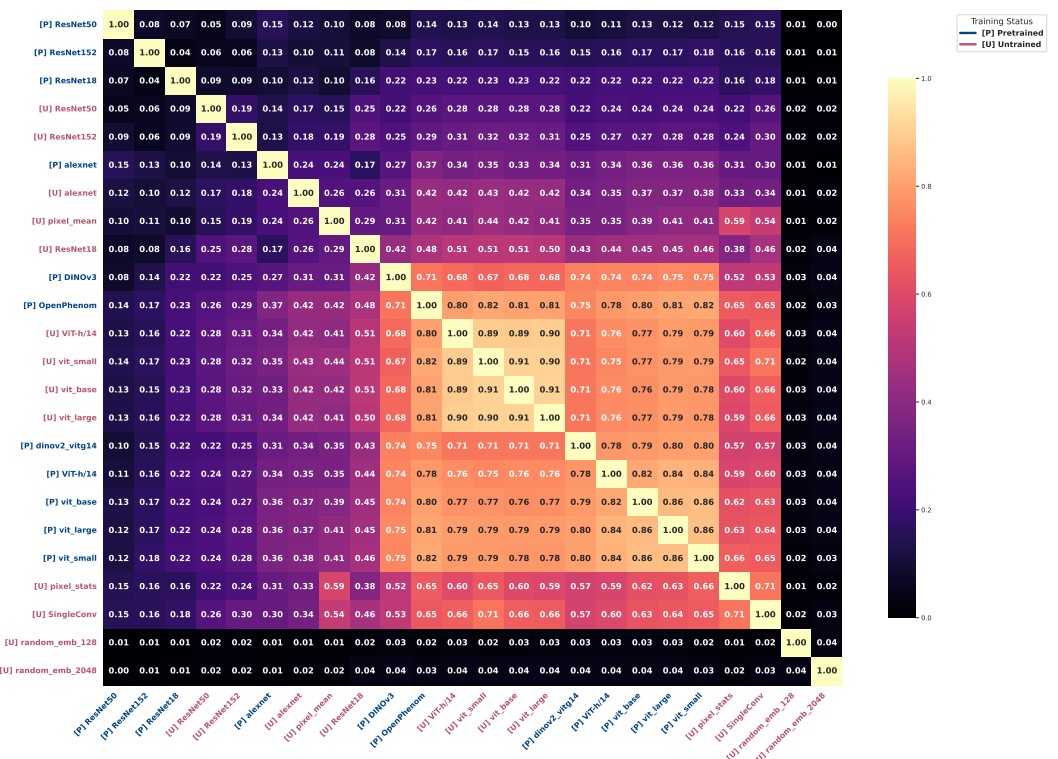

Figure 21: **Representational Similarity Analysis (RSA) across model architectures.** The heatmap displays the pairwise similarity between model embeddings calculated using Spearman's rank correlation coefficient. Models are hierarchically clustered to reveal functional groupings. Labels prefixed with **[P]** (blue) denote pretrained models, while **[U]** (red) indicates untrained/random initializations.

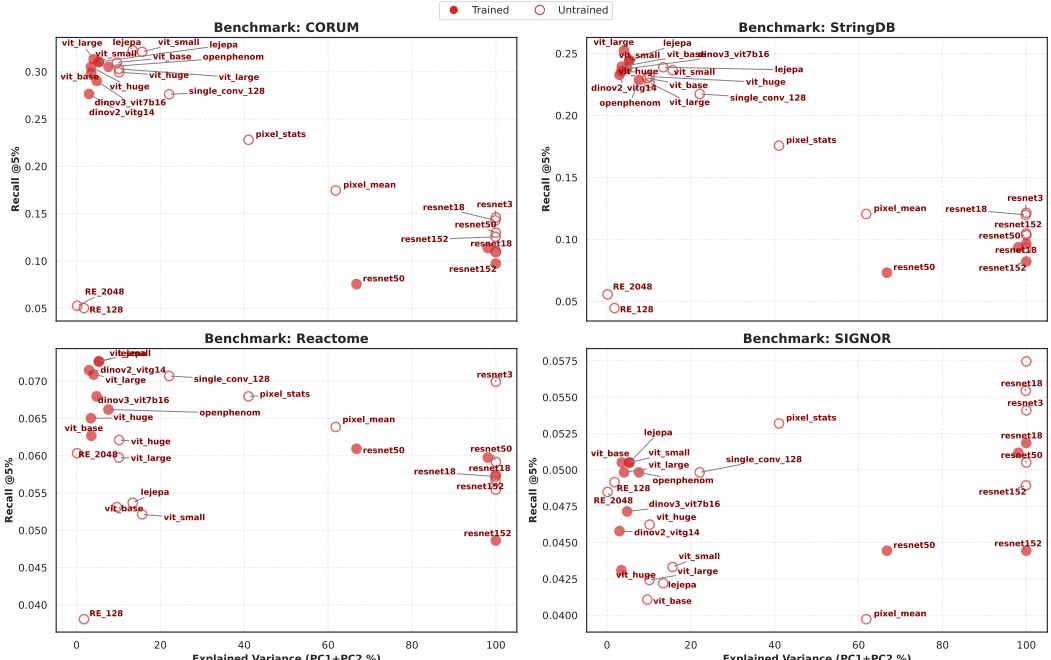

Figure 22: PCA Variance Explained of RxRx3-Core vs. Recall @5% across various datasets (CORUM, StringDB, Reactome and SIGNOR).

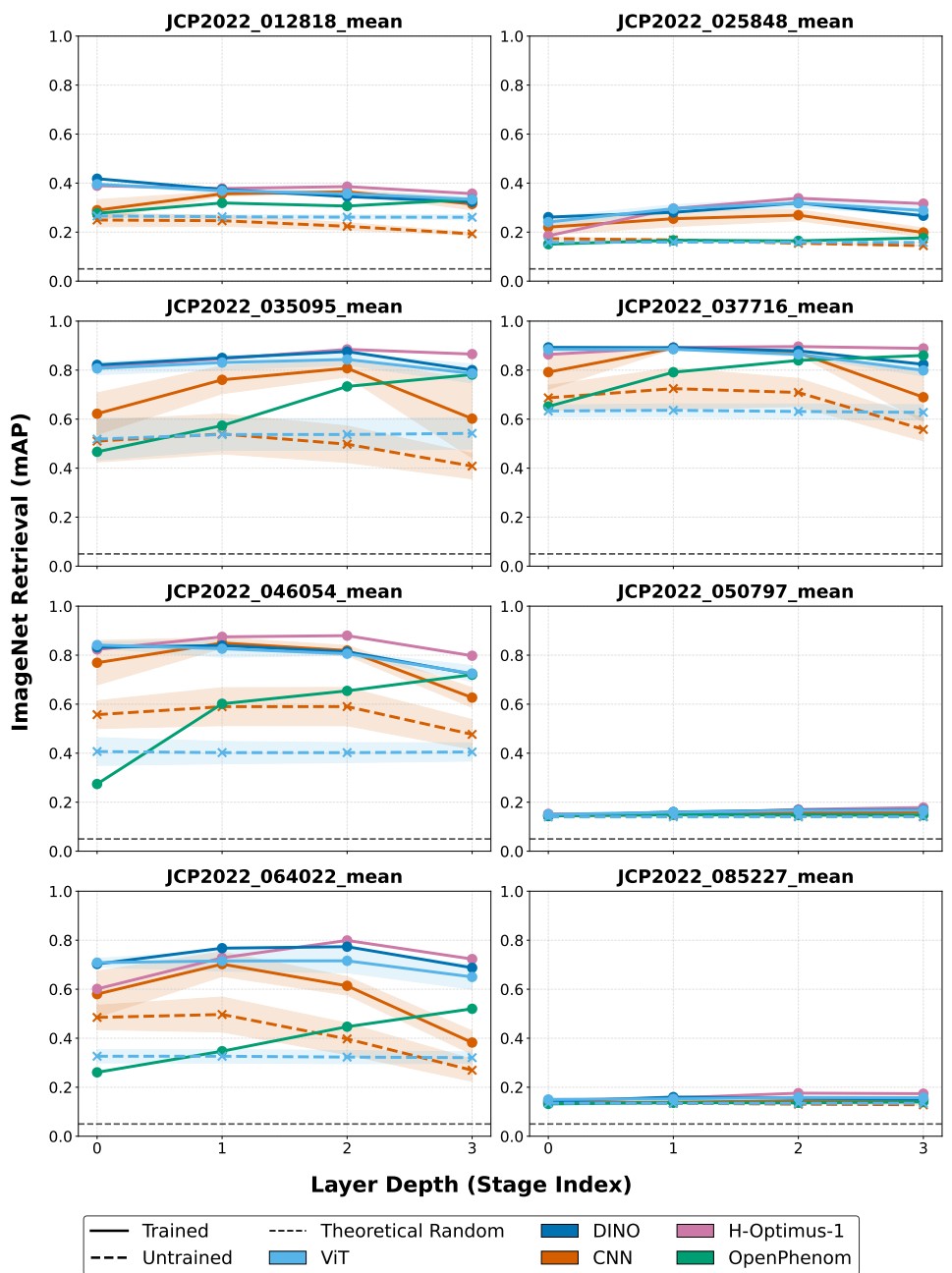

Figure 23: **Evolution of representation quality across intermediate layers for JUMP-CP compounds.** Performance is evaluated as Mean Average Precision (mAP) for ImageNet retrieval across eight distinct positive control compound. Each panel displays the mAP score as a function of the network intermediate layers (Layer Depth). Solid lines denote pretrained models, while dashed lines indicate untrained models; shaded areas represent variability across a given architecture. The dashed horizontal line in each plot represents the theoretical random baseline (0.125).

# I  UPSCALED MAIN TEXT FIGURES.

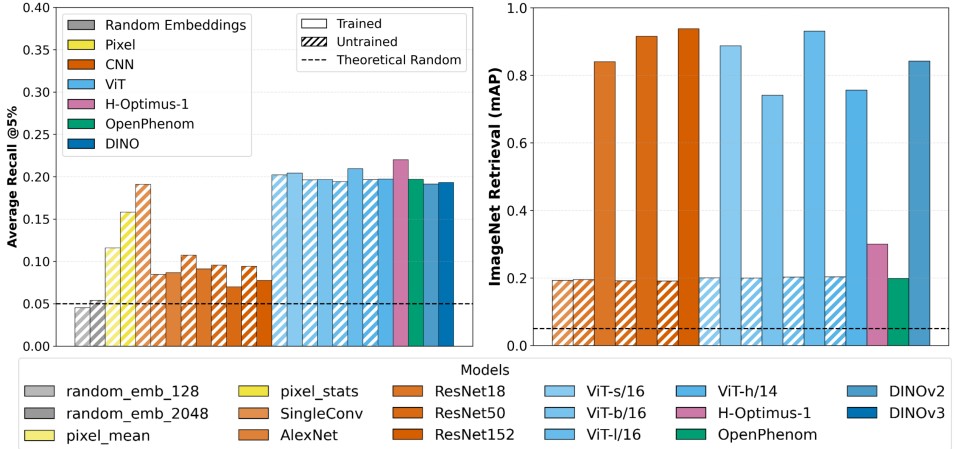

(a) Comparison between trained and untrained models on natural images and Cell Painting data from RxRx3-core. *Right*: kNN top-1 accuracy on ImageNet-1k across models. *Left*: gene–gene retrieval performance on RxRx3-core (Recall@5%). Mean recall across all 5 literature datasets.

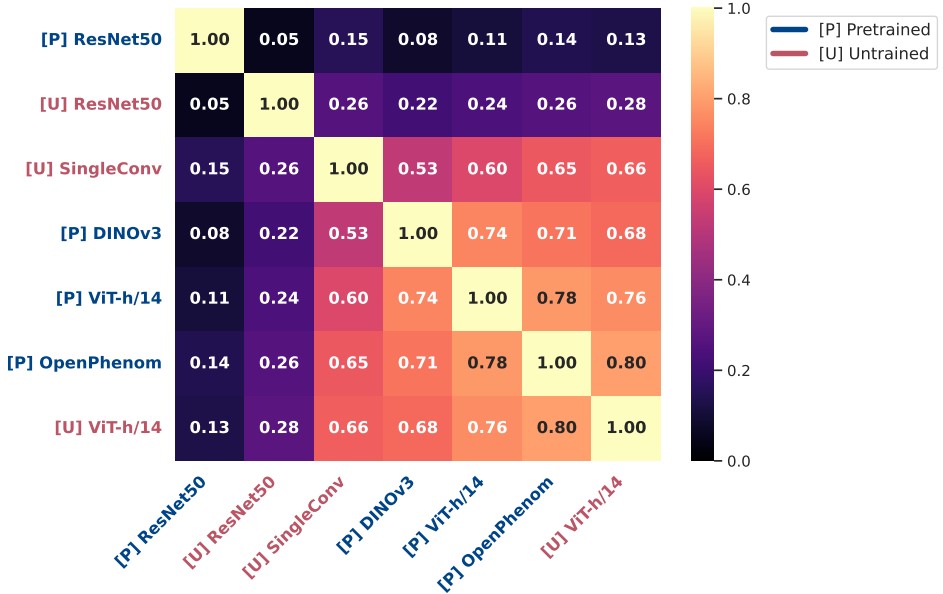

(b) Spearman correlations between gene–gene similarity rankings induced by different representations on RxRx3-core genes. Models are hierarchically clustered to reveal functional groupings.

Figure 24: **(Figure 1) Comparison between trained and untrained models.** (a) Task-dependent metrics on natural images and Cell Painting data. (b) Correlations between predictions on Cell Painting data.

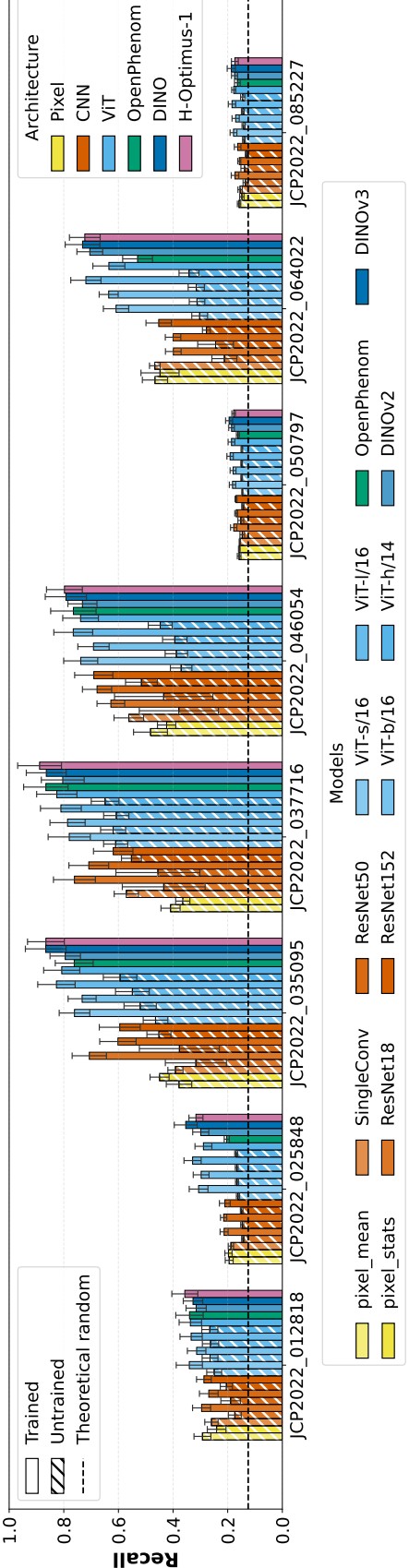

Figure 25: **(Figure 3) Mean average precision (mAP) per compound on the JUMP-CP benchmark.** Bar plots show mAP scores for eight positive-control compounds and the mean across compounds. Each bar corresponds to a model configuration, grouped and colored by architecture family. Solid bars denote pretrained models, hatched bars denote untrained models, and pixel-based baselines are included for reference. Error bars indicate variability across evaluation folds.

