# OpenReview forum: "Deep Learning for BioImaging: What Are We Learning?"
_ICLR.cc/2026/Workshop/FM4Science — ICLR 2026 Workshop FM4Science Poster_

### Official Review · Reviewer_69fw · 2026-02-20
**Limited insights with impacts of representation learning on biomedical image**

**Rating:** 4
**Confidence:** 5

**Review:**

The authors investigated current overview and limits in representation learning on biomedical image. The main contribution of this paper included 1.) current model performance  of microscopy benchmarks were gerenerally lacking capability to catch biological  abstraction. 2.) regardless model architecture and model embedding outputs, the performance of models failed to catch biological semantics. Although diverse models were investigated and tested on different benchmarks, this manuscript suffered from lack of technical novelty. It is unclear motivation why the authors spent efforts on investigation of untrained pre-trained models during investigation, which yielded established statements already.
Below are my thoughts:
1. Why graph representation learning was not investigated given the fact that cell graphs were constructed？
2. Fig5 delivered little useful information. The authors may consider remove from the manuscript.
3. I assume this paper audience is researchers in biomedical field. Therefore, I suggest discussion the tradeoff of cost and performance with them instead of only focusing on model performance.

---

### Official Review · Reviewer_mx9z · 2026-02-23
**Review of the paper "Deep Learning for BioImaging: What Are We Learning?"**

**Rating:** 7
**Confidence:** 3

**Review:**

**Summary** - The paper investigates a fundamental question of whether current deep learning and foundation models actually learn biologically meaningful representations in microscopy imaging. The authors argue that progress in microscopy representation learning may be overstated because benchmarks can reward dataset shortcuts rather than biological abstraction. The work introduces diagnostic baselines designed to test what information is truly needed to achieve strong benchmark performance, and evaluate models across two biological scales - Cell culture microscopy and Tissue imaging. Representations are evaluated using frozen embeddings and retrieval-style metrics.

**Strengths**-
1. The work targets a core issue in scientific ML, and highly relevant to foundation models in biology.
2. The use of untrained networks, pixel baselines, structure-only models are well designed.
3. Strong experiments with interesting insights on architectural bias and relationship between dataset structure and metrics

**Weaknesses** -
1. The evaluation relies on the same benchmarks that the paper critiques. While the work argues that existing benchmarks may not faithfully measure biological understanding, conclusions are still derived primarily from performance on those benchmarks. This creates a degree of evaluation circularity and makes it difficult to disentangle whether observed effects reflect shortcomings of models or limitations of the benchmark design itself.
2. Moreover, the empirical evaluation relies primarily on balanced accuracy, which may be insufficient for assessing clinical utility. In medical imaging settings with highly imbalanced disease prevalence, recall (sensitivity) is typically the critical metric due to the high cost of false negatives. Balanced accuracy implicitly weights sensitivity and specificity equally and may therefore blur clinically relevant failure modes. The paper would benefit from reporting recall-oriented metrics (e.g., sensitivity at fixed specificity, PR-AUC) to better reflect real-world deployment scenarios.

---

### Meta-Review · Area_Chair_ZE9B · 2026-03-02

**Recommendation:** Accept (Poster)
**Confidence:** 3

**Metareview:**

This submission has received two reviews. One with a "clear accept" and one with "Ok but not good enough".

After reading the reviews, I recommend this paper for "acceptance" and ask the authors to consider implementing the feedback given by both reviewers into the camera-ready version of the paper.

---

### Decision · Program_Chairs · 2026-03-03

Accept (Poster)